# Ecological memory preserves phage resistance mechanisms in bacteria

Antun Skanata[1] & Edo Kussell [1,2✉]

Bacterial defenses against phage, which include CRISPR-mediated immunity and other mechanisms, can carry substantial growth rate costs and can be rapidly lost when pathogens are eliminated. How bacteria preserve their molecular defenses despite their costs, in the face of variable pathogen levels and inter-strain competition, remains a major unsolved problem in evolutionary biology. Here, we present a multilevel model that incorporates biophysics of molecular binding, host-pathogen population dynamics, and ecological dynamics across a large number of independent territories. Using techniques of game theory and non-linear dynamical systems, we show that by maintaining a non-zero failure rate of defenses, hosts sustain sufficient levels of pathogen within an ecology to select against loss of the defense. This resistance switching strategy is evolutionarily stable, and provides a powerful evolutionary mechanism that maintains host-pathogen interactions, selects against cheater strains that avoid the costs of immunity, and enables co-evolutionary dynamics in a wide range of systems.

[1] Department of Biology & Center for Genomics and Systems Biology, New York University, New York, NY 10003, USA. [2] Department of Physics, New York University, New York, NY 10003, USA. ✉email: edo.kussell@nyu.edu

Survival of species in the presence of pathogens requires effective defense mechanisms, which exist in a wide range of biological systems[1–4]. Host-pathogen interactions are subject to the availability of susceptible hosts that sustain a viable pool of pathogens, while defense mechanisms are under evolutionary pressure to reduce or eliminate the ability of pathogens to proliferate. Once a threat is removed so is the pressure to preserve the relevant defense mechanism which may carry significant fitness costs[5–7]. In the absence of defenses, pathogens can reemerge and wipe out a host population. The selective forces that drive the defense mechanisms to become highly efficient may thus eventually lead the host to extinction. How species avoid this inherent fragility of their defenses is not presently understood[8].

Bacteria and their pathogens, the bacteriophages, present a powerful system to study this question. A bacteriophage infects a bacterial cell by attaching to the cell surface and injecting its genetic material, then replicates and assembles phage particles, which are released upon cell lysis (Fig. 1a). Bacteriophage resistance mechanisms[1] exhibit a great deal of variety across two major classes: (i) preventative defenses, which operate by loss, modification, or blocking of phage receptors[9–12], and (ii) immune defenses, such as restriction-modification systems[13] and CRISPR[14] which cleave phage DNA after it enters the host. Costs of resistance mechanisms[15] have been demonstrated in different preventative[16–18] and immune[19,20] defense systems. The diversity of phage-host systems in terms of routes of infection and modes of resistance points to strong evolutionary pressures favoring the emergence and maintenance of resistance[2,21,22].

A well-studied example of a preventative defense is found in the host-pathogen interaction of *Escherichia coli* and the phage $\lambda$, which attaches to the host cell through the maltoporin LamB, an outer membrane protein. Preventative resistance is conferred through mutations in *lamB* or in the activator of the maltose regulon, *malT*, which result in loss of *lamB* expression, growth defects on maltose[17,23], and resistance to $\lambda$ phage. A subset of *malT* mutations, which involve duplications or insertions, revert at frequencies of $10^{-5}$–$10^{-4}$ per division, leading to spontaneous induction of a phage-sensitive phenotype in an otherwise resistant population[24,25]. In these 'resistance switching' strains, phage persist at low frequencies, while in strains that do not switch off their resistance (e.g. *lamB* mutants) phage cannot be sustained

and go extinct[24]. In the absence of phage, resistant mutants can be outcompeted by sensitive strains which do not bear the cost of resistance, leading to loss of resistance in a population.

To study the maintenance of defenses, we model phage-bacteria interactions at the molecular, population, and ecological levels. At the molecular level, we consider the kinetics of phage-receptor binding and phage absorption into the cell, and from these we derive the dependence of the infection rate on phage and bacteria concentrations and molecular parameters. The functional form of the infection rate is then used to build a population dynamics model of sensitive and resistant bacteria growing in the presence of phage. We determine the fixed points of the dynamical system, which correspond to steady-state solutions for which the composition of the bacteria and phage population remains constant in time, as well as cases in which a limit cycle exists. Fixed points of this single-population model can be stable or unstable to perturbations such as migration of a new bacterial strain into the population. We show that in a single population, successive replacements of strains (e.g. sensitive, resistant, or resistance switching) can occur without leading to a stable long-term outcome. This motivates us to consider ecological dynamics across a large set of patches, each of which corresponds to a local population, and where co-invasion of different strains into unoccupied patches drives changes in patch composition across an ecology. We construct the ecological model from the single-population model, by letting each fixed point of the population dynamics correspond to a distinct patch type. At the ecological level, we show that the time evolution of patch frequencies is given by a replicator equation. This enables us to apply the tools of game theory to analyze the long-term outcomes of the ecological dynamics.

In this work we show that spontaneous loss of resistance in single cells – which enables phage to persist in the host's environment – protects the resistance mechanism from eventual loss at the ecological level. Specifically, we demonstrate that resistance switching is an evolutionarily stable strategy (ESS) that can be naturally evolved. Beyond this, our analysis shows that differences in defense mechanisms that manifest at the molecular level can have a large impact on population dynamics. In particular, for immune defenses the absorption of phage by resistant cells enables coexistence with sensitive strains which do not pay the cost of resistance. We show that such immune cheaters can be selected against using the resistance switching strategy.

## Results

### Multilevel model: Molecular, cellular, population, and ecological levels.
We introduce a multilevel model of bacteria and phage dynamics with: (i) molecular and cellular levels, which model the phage receptor kinetics and yield the functional form of the infection rate; (ii) population level, which uses the infection rate in a system of differential equations describing the population dynamics within a single patch; and (iii) ecological level, which models competitions among a set of bacterial strains across a large number of patches. Table 1 summarizes the key notation used at different levels of the model. The specific values of the molecular parameters (Fig. 1) determine the set of fixed points of the population dynamics equations (Figs. 2, 3). These fixed points in turn dictate the different kinds of migration and invasion dynamics that can occur in a single patch (Fig. 4). Analysis of fixed points for CRISPR systems is shown in Fig. 5. Modeling and analysis at the ecological level using game theory is shown schematically in Fig. 6.

At the molecular and cellular levels, we consider the coarse-grained mechanism of infection by a phage freely diffusing onto a host cell[26–28] (Fig. 1a). We consider a phage-receptor binding

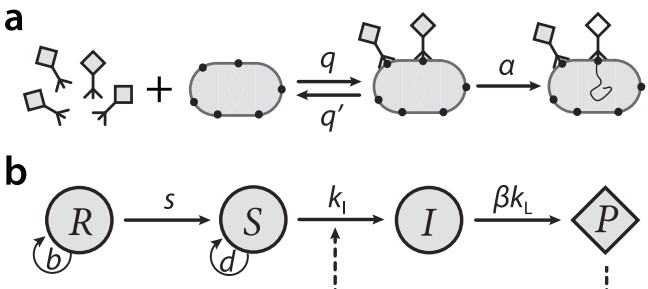

**Fig. 1 Phage-host molecular interactions and population dynamics.**
(**a**) Molecular interactions. The phage diffuses onto a host that expresses phage receptors (solid circles), binds reversibly to a receptor and infects the cell by injecting its DNA. Reaction rates are shown on arrows; the phage binding and unbinding rates $q$ and $q'$ and phage absorption rate $\alpha$ are defined per receptor. (**b**) Population dynamics for a preventative defense model. The resistant phenotype $R$ generates phage-sensitive cells $S$ at rate $s$, on which the phage $P$ grows. $I$ labels cells infected by phage. Arrows indicate rate constants for each type of transition; the circular arrows indicate the growth rates of $R$ and $S$ cells (respectively, $b$ and $d$) with $d - b > 0$ being the cost of resistance. For immune defenses, all phenotypes express receptors and absorb phage, but only sensitive cells transition to infection and lysis.

**Table 1 Notation Guide. Key symbols and their meaning for each stage of the multi-level model.**

| Symbol | Definition |
|---|---|
| **Molecular and cellular levels** | |
| $r_f$, $r_b$ | Receptor concentration (free, bound) |
| $n_r$ | Number of receptors per cell (average) |
| $q$, $q'$ | Receptor-phage binding, unbinding rates |
| $\alpha$ | Phage absorption (injection) rate |
| $K_m$ | Michaelis constant for receptor-phage binding |
| $k_I(t)$ | Infection rate of sensitive cell |
| **Population level** | |
| $R(t)$, $S(t)$, $I(t)$ | Phenotype concentration (resistant, sensitive, infected) |
| $A(t)$ | Total concentration of phage-absorbing phenotypes |
| $P(t)$ | Phage concentration |
| $\lambda(t)$ | Dilution rate |
| $d$, $b$ | Growth rate of sensitive, resistant cells |
| $s$ | Switching rate from resistant to sensitive phenotype |
| $\beta$ | Phage burst size (average) |
| $k_L$ | Lysis rate of infected cell |
| **Ecological level** | |
| **S** | Sensitive strain (phenotypes: S,I) |
| **R**$_O$ | Resistant strain (phenotype: R) |
| **R**$_s$ | Resistance switching strain (phenotypes: R, S, I) |
| $x_i(t)$ | Frequency of strain $i$ |
| $f_{ij}$ | Fitness difference between strains $i$ and $j$ |
| $c$ | Patch clearing rate |

reaction

$$r_f + P_f \underset{q'}{\overset{q}{\rightleftharpoons}} r_b \xrightarrow{\alpha} r_f \tag{1}$$

where $r_f$ and $r_b$ denote concentrations of free and bound receptor, $P_f$ and $P_b$ denote concentrations of free and bound phage, and we have $P_b = r_b$. Reversible binding/unbinding of phage occurs at rates $q$ and $q'$ per receptor. Irreversible injection of phage genetic material into the cell occurs at rate $\alpha$, which we refer to as the phage absorption rate. We denote by $A$ the concentration of phage-absorbing host cells, and by $n_r$ the average number of receptors per cell.

We define the rate of infection per cell to be $k_I(t)$, and note that the total rate of infection in the population is given by $k_I(t)A = \alpha r_b$. In a quasi-steady-state approximation the concentration $r_b$ is constant over timescales $1/\alpha$, and we solve a quadratic equation to obtain its dependence on the total concentration of receptors, $r \equiv r_f + r_b$, and phage $P \equiv P_f + P_b = P_f + r_b$. We then use the fact that $r = n_r A$ to obtain the general form of the infection rate per cell as

$$k_I(t) = \frac{\alpha n_r P}{K_m + n_r A + P} , \tag{2}$$

where $K_m \equiv (q' + \alpha)/q$. Given the molecular parameters of the receptor, which determine $K_m$, the phage-bacteria interaction can exhibit different dependencies. For low receptor binding affinity, $K_m \gg n_r A, P$, we obtain the total infection rate $k_I(t)A = (\alpha n_r/K_m)PA$, which corresponds to a Lotka-Volterra interaction. For high receptor binding affinity, $K_m \ll n_r A, P$, we can omit $K_m$ in the denominator of Eq. (2). This yields a hyperbolic dependence on $P/(n_r A)$, the multiplicity of infection (MOI) per receptor. Experimentally determined values of $K_m$ are in the high binding affinity limit (see Methods, 'Phage-receptor binding and infection rate').

At the population level, we model a single population of phage ($P$) and bacteria that express a resistant phenotype ($R$) or a sensitive phenotype ($S$) (see Fig. 1b), according to the following equations

$$\begin{aligned}
\dot{R}(t) &= (b - s)R(t) - \lambda(t)R(t), \\
\dot{S}(t) &= d\, S(t) + sR(t) - k_I(t)S(t) - \lambda(t)S(t), \\
\dot{I}(t) &= k_I(t)S(t) - k_L I(t) - \lambda(t)I(t), \\
\dot{P}(t) &= -k_I(t)A(t) + \beta k_L I(t) - \lambda(t)P(t) ,
\end{aligned} \tag{3}$$

where $R$, $S$, $I$, and $P$ are expressed in concentration units. Phage infect sensitive cells at rate $k_I(t)$ determined above (Eq. (2)) generating infected cells ($I$), and these lyse at rate $k_L$ to produce $\beta$ new phage particles; $\beta k_L$ is the phage burst rate, the overall rate at which phage particles are released into the environment. Resistance switching is modeled as spontaneous conversion of $R$ cells into $S$ cells at rate $s$, and the cost of resistance, $d - b > 0$, is given by the difference in growth rates of sensitive and resistant cells. $A$ is the total number of phage-absorbing hosts: for preventative defenses $A = S + I$, as only sensitive and infected cells absorb phage, while for immune defense systems $A = R + S + I$. The model can also include phage decay by adding a term $-\delta P$ to the last equation (see Methods, 'Phage decay'). The dilution rate $\lambda(t)$ is chosen to implement different types of population growth control. We present results for growth in rich media with feedback dilution (turbidostat growth) in the main text, or in a nutrient-limited environment with constant dilution (chemostat growth) in Supplementary Note 1; the specific growth modality does not impact the major outcomes. We analyze Eq. (3) to determine fixed points, which correspond to steady-state population compositions, and their stability to perturbations (see Methods, 'Linear stability analysis').

At the ecological level, we model the dynamics of migration and invasion that take place in the setting of a large number of patches, each of which consists of a single population as modeled above. Each patch is taken to correspond to a fixed point (or more generally to any invariant set, e.g. a periodic orbit) of the single population dynamics equations. We index the possible patch types by $i$, which corresponds to all possible fixed points, stable or unstable, for a given choice of model parameters, and denote by $x_i$ the frequency of each patch type, where $0 \le x_i \le 1$ and $\sum_i x_i = 1$. Patch clearing events, which occur with rate $c$ per patch per unit time, clear a patch of its inhabitants and enable invasion by strains from other patches. We assume that cleared patches are rapidly invaded at rate $m$, such that patch clearing is the rate limiting step to initial patch invasion, i.e. $c \ll m$. When invasions occur, they bring in a small inoculum from a patch of type $i$ with probability $x_i$. Patch clearing and migration mimic natural turnover events that occur e.g. in the gut due to peristalsis, in soil microenvironments due to physical perturbations, or during host-to-host transmission in epidemiology.

In the initial stages of patch invasion, we postulate that it takes a time $\tau_{est}$ for an invading type to establish on the patch and thereafter exclude other invaders. If no further migration events occur during time $\tau_{est}$ after the initial invasion, the patch will become type $i$. However, since the patch clearing rate is independent of patch type, and invading types are chosen randomly according to the distribution $x_i$, there will be no net change of $x_i$ due to single invasion events. Thus, the dynamics of $x_i$ are determined by multiple invasion events, which occur with total rate $c(1 - e^{-m\tau_{est}}) \approx c\, m\tau_{est} + c\, \mathcal{O}[(m\tau_{est})^2]$. We can therefore disregard invasion events from three or more patches if $m\tau_{est} \ll 1$, e.g. for short establishment time and/or rare immigration, and consider the dynamics due to co-invasion events, i.e. competitions between invaders from exactly two

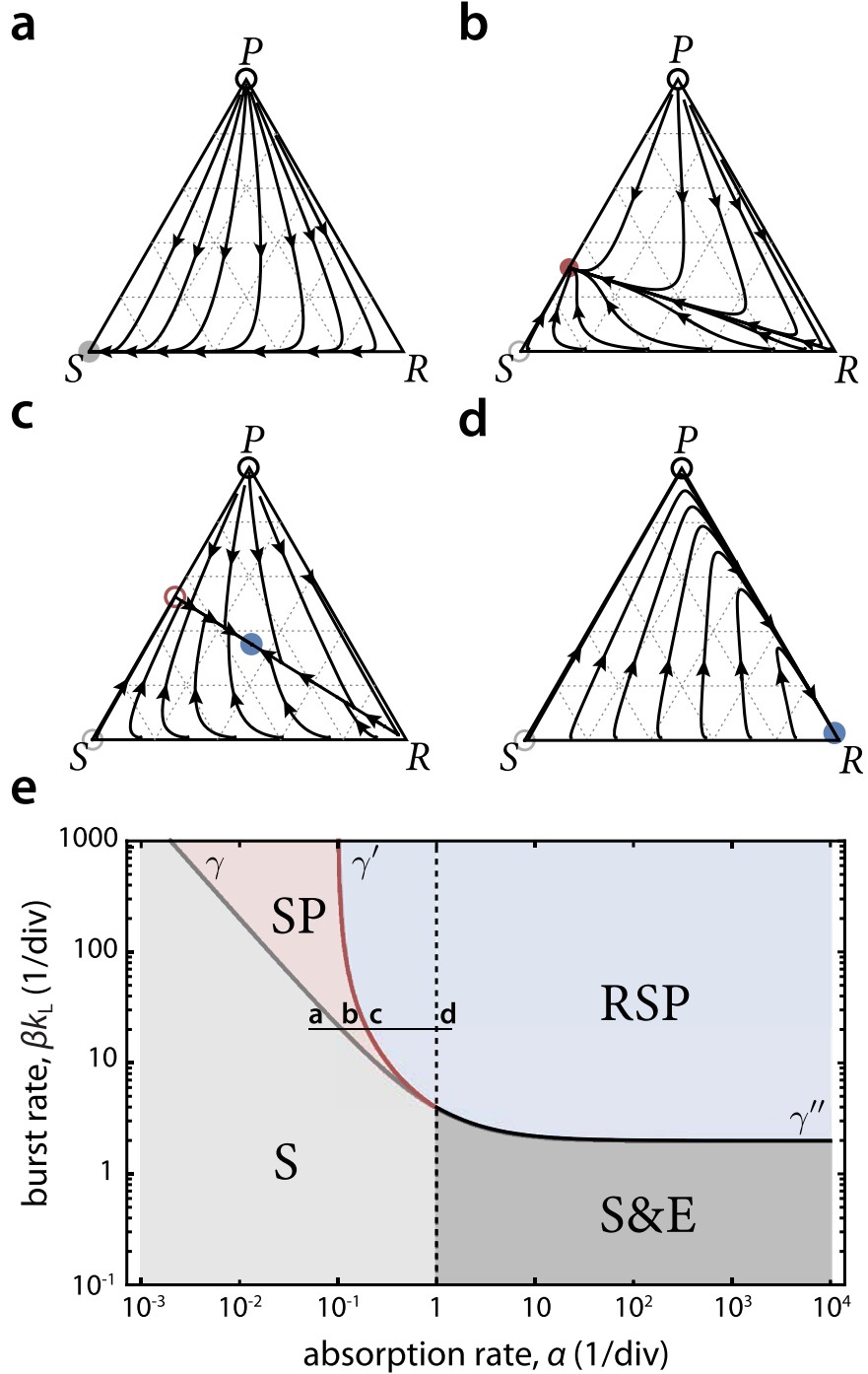

**Fig. 2 Stable phases for preventative defenses.** (**a–d**) Flow diagrams of Eq. (3), plotting frequencies of the resistant ($R$) phenotype along the bottom edge, sensitive ($S$) phenotype along the left edge and phage and infected (as $P + I$) along the right edge in a ternary plot. Empty (filled) circles - unstable (stable) fixed points. Color of the fixed point matches the phase displayed in the phase diagram below. Host extinction at the top corner is represented with a black empty circle. (**e**) Phase diagram for preventative defenses. The S phase (gray region) carries only the $S$ phenotype, the SP phase (pink region) carries the $S$ phenotype and phage, the RSP phase (blue region) carries all phenotypes and phage, the S&E phase (dark gray region) corresponds to bistability between phase S or host extinction E. The curves $\gamma$ (gray), $\gamma'$ (dark red) and $\gamma''$ (black) denote locations of bifurcations. Letters a-d indicate the locations corresponding to the flow diagrams above. Rates are expressed in units of per cell division time (1/div), with $d = 1$, $b = 0.9$, $k_L = 1$ and $s = 10^{-4}$. Results are shown for high phage-receptor binding affinity ($K_m \to 0$) and minimal sensitivity ($n_r = 1$). See Methods, `Stability analysis at low binding affinity' for $K_m > 0$ and Supplementary Figs. 7–9 for dependence on $n_r$, $b$ and $k_L$.

patches. We note that co-invasion does not require simultaneous arrival of competitors, as two strains can arrive within time $\tau_{est}$ of each other.

Co-invasion events from patch types $i$ and $j$ occur with rate $2c \, m\tau_{est}x_ix_j$, for $i \neq j$, and rate $c \, m\tau_{est}x_i^2$ for $i = j$. If $i = j$, the types

are identical resulting in a patch of type $i$. If $i \neq j$, competition occurs and resolves itself over a timescale that is determined by the fitness difference $f_{ij} \equiv |f_i - f_j|$ between the two types, where $f_i$ is the fitness of type $i$. When $f_i > f_j$, the competition will resolve in favor of type $i$, provided that the competition is over before the

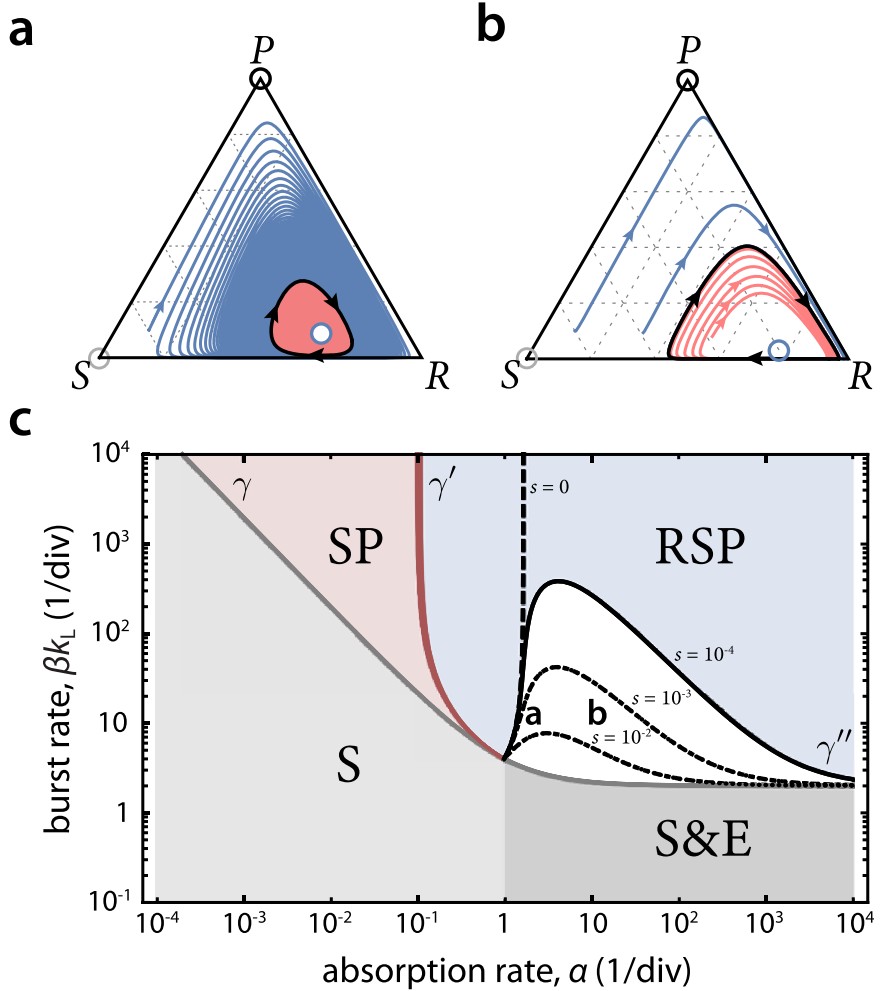

**Fig. 3 Stable phases for immune defenses.** (**a,b**) Flow diagrams of Eq. (3) applied to immune defenses show periodic orbits occurring in the RSP region of the phase diagram shown in panel c. Blue and pink curves show trajectories that approach the orbits from two directions. The blue empty circle marks the location of the unstable interior fixed point. (**c**) Phase diagram for immune defenses. The S, SP, RSP and S&E phases are defined as in Fig. 2. Curves $\gamma$ (gray), $\gamma'$ (dark red) and $\gamma''$ (black) denote locations of bifurcations. The white region bounded by $\gamma''$ (shown for $s = 10^{-4}$) contains no stable fixed points and exhibits a stable limit cycle. Dotted black curves denote the $\gamma''$ curve for higher switching rates; the dashed black curve corresponds to $\gamma''$ for $s = 0$. Letters a and b indicate approximate locations corresponding to the flow diagrams above (using $s = 10^{-3}$). Remaining parameters are as in Fig. 2.

next patch clearing event. If patch clearing occurs before resolution, then neither type makes any gains. The probability of successful resolution in favor of type $i$ is thus $p_{ij} \equiv f_{ij}/(f_{ij} + c)$, if $f_i > f_j$, or $p_{ij} = 0$ if $f_i < f_j$. The total rate of co-invasions of $i \neq j$ successfully resolving in favor of $i$ is $2c \, m\tau_{est}p_{ij}x_ix_j$. This yields the patch frequency dynamics

$$\dot{x}_i = c \, m\tau_{est}x_i\left(x_i + 2\sum_{j\neq i} p_{ij}x_j - \bar{F}\right) \quad (4)$$

where $\bar{F} \equiv \sum_i x_i^2 + 2\sum_{i,j:j\neq i}p_{ij}x_ix_j$. The above equation takes the form known in game theory as the replicator equation,

$$\dot{x}_i = x_i\left[(\phi\vec{x})_i - \vec{x}\cdot\phi\vec{x}\right], \quad (5)$$

where the payoff matrix $\phi$ is given by $\phi_{ij} \equiv cm\tau_{est}[\delta_{ij} + (1 - \delta_{ij})2p_{ij}]$. Since the prefactor $cm\tau_{est}$ sets the overall timescale but does not otherwise impact the dynamics, we set it equal to 1 by rescaling the time unit. The equilibria of patch frequency dynamics can therefore be analyzed using game theory, i.e. through the identification of Nash equilibria and evolutionarily stable strategies.

**Stable phases of a phage and host population.** The state of a single population of phage and bacteria can be visualized as a ternary plot in which each corner corresponds to a monomorphic population composed entirely of sensitive cells, resistant cells, or phage particles (Fig. 2a and Supplementary Fig. 1). Flow lines within the diagram show the solutions of the model starting from different initial conditions. Points in the interior of the triangle represent different number compositions of host cell phenotypes and phage, using total counts of cells and phage for normalization; one can alternatively plot the composition as biomass fractions, which is a one-to-one transformation of the ternary plot that preserves all topological features including fixed points, trajectory structure, and stability (see Methods, 'Population structure and control of total biomass'). We present results separately for the preventative and immune defense models below.

Representative flow diagrams for the preventative defense model, in which resistant cells do not absorb the phage, are shown in Fig. 2a-d, where each panel corresponds to a different phage absorption rate $\alpha$ and burst rate $\beta k_L$. Across all possible combinations of phage parameters, there are four possible long-term outcomes of the dynamics which correspond to fixed points

of the model equations: (i) fixation of S [S phase], (ii) coexistence of S and P [SP phase], (iii) coexistence of R, S, and P [RSP phase] and (iv) host extinction [E phase].

Depending on the phage parameters, one or more of these phases may be stable to small perturbations of the population composition. The stable phases are shown as distinct regions in the space of phage parameters (Fig. 2e), separated by curves $\gamma$ and $\gamma'$, which correspond to transcritical bifurcations of the dynamical system (see Methods, 'Linear stability analysis', and Supplementary Fig. 2). Curve $\gamma''$ where the RSP fixed point becomes unstable is obtained numerically. There also exist regions where two phases are stable (for $\alpha > d/n_r$), and which phase is observed depends on initial conditions; these include the S&E bistable region and a narrow region of RSP&S bistability (Supplementary Fig. 3). The phase diagram for a model that lacks the resistance phenotype is shown in Supplementary Fig. 4 and analyzed in Supplementary Note 2.

In the S phase, which is the unique stable phase for $\alpha < d/n_r$ and $\beta k_L < \gamma$, the resistant phenotype and phage will be outgrown by the sensitive phenotype (Fig. 2a). As $\alpha$ increases along the thin black horizontal line in Fig. 2e, it crosses the curve $\gamma$, where the phage can coexist with the host in the SP phase (Fig. 2b). In this phase, the growth rate of S, while reduced by the phage, is still larger than the growth rate of R, and resistance cannot establish.

With further increase in $\alpha$, the growth rate of S cells decreases until it equals the growth rate of R at the location of $\gamma'$, where the system transitions to the RSP phase (Fig. 2c). Beyond $\gamma'$, the frequency of R at the RSP stable fixed point increases with $\alpha$, while the frequencies of S, I and P decrease (Fig. 2d).

In the RSP phase, any amount of non-zero switching ($s > 0$) will generate S on which phage can grow, and promote phage presence in the host population. Interestingly, in this phase a phage that infects at a higher rate will be present at lower frequency, because resistance will have a higher selective advantage in the presence of a stronger pathogen (Supplementary Fig. 2). In the absence of switching ($s = 0$), the RSP phase reduces to a pure resistant population. We note that the model does not distinguish between mechanisms underlying resistance switching, e.g. if both mutations and epigenetic switching are present, these would both contribute to the effective value of $s$ in the model. We additionally show that resistance switching allows distinct phage strains with different parameters to coexist within the same host population (Supplementary Fig. 5 & Supplementary Note 3).

Representative flows and the phase diagram for immune defenses, in which all phenotypes absorb the phage, are shown in Fig. 3a–c. Absorption of phage by the resistant cells directly couples phage and resistant subpopulations in Eq. (3) and generates a region where none of the four possible fixed points are

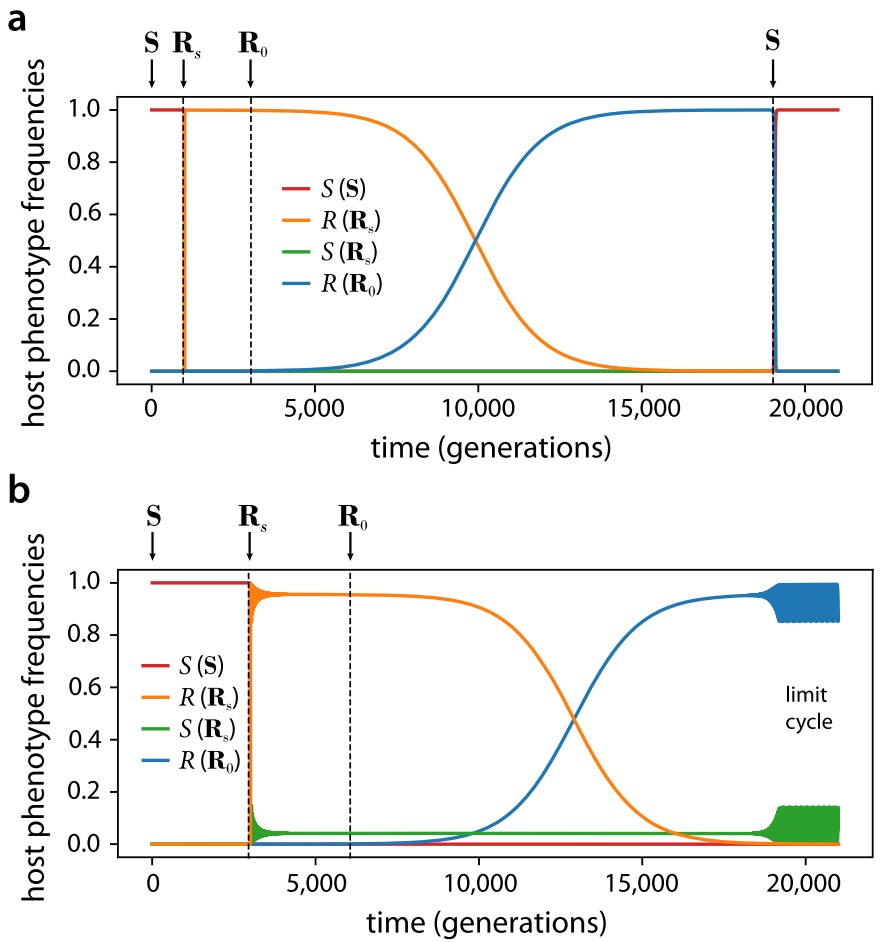

**Fig. 4 Invasion dynamics for (a) preventative and (b) immune defense systems showing phenotype frequencies in the host population.** Arrows indicate times at which an invading strain is added to the population. We label by $R(\mathbf{R}_s)$ and $R(\mathbf{R}_O)$ the resistant phenotype of each strain separately, and similarly by $S(\mathbf{R}_s)$ and $S(\mathbf{S})$ the sensitive phenotypes. The resistance switching strain $\mathbf{R}_s$ outcompetes the sensitive strain $\mathbf{S}$ as it carries phage which lyse sensitive cells. In (a), $\mathbf{R}_O$ invades and eventually replaces $\mathbf{R}_s$, and can then be invaded by $\mathbf{S}$. In (b), $\mathbf{R}_O$ invades $\mathbf{R}_s$, replacing the switching $R$ phenotype while coexisting with the $S$ phenotype of the $\mathbf{R}_s$ strain and its phage. The RSP fixed point, which was stable for the $\mathbf{R}_s$ strain, is unstable for the $\mathbf{R}_O$ strain and the dynamics transitions to a stable limit cycle. The model uses turbidostat control with $s = 10^{-3}$, $\alpha = 10$, $\beta = 50$; all remaining parameters are as in Fig. 2.

stable. In this region, stable limit cycles are possible in which $R$, $S$, and $P$ levels oscillate periodically (Fig. 3c, white region); this region is therefore part of the RSP phase. In Fig. 3a and b we show two such orbits, one for the dynamics located near the edge of that region and one located further inside the region. The limiting orbit in Fig. 3b passes extremely close to the $P = 0$ boundary, a behavior that in finite systems would eventually lead to loss of phage and collapse to a stable S fixed point. For higher switching rates, such orbits are pulled away from the boundary and toward the interior of the simplex (Supplementary Fig. 6).

A Hopf bifurcation curve $\gamma''$ separates the stable and unstable fixed points of the RSP phase, and is shown in Fig. 3c for different values of $s > 0$ (solid and dotted black curves) or for $s = 0$ (dashed black curve). As $s$ increases from zero, $\gamma''$ confines the region of periodic dynamics to lower phage burst rates, while the $\gamma'$ curve shifts (slightly) to higher $\beta k_L$ across the thickness of the dark red curve in Fig. 3c.

**Invasion dynamics in a single patch**. We now examine invasions in a single patch or territory occupied by distinct host strain

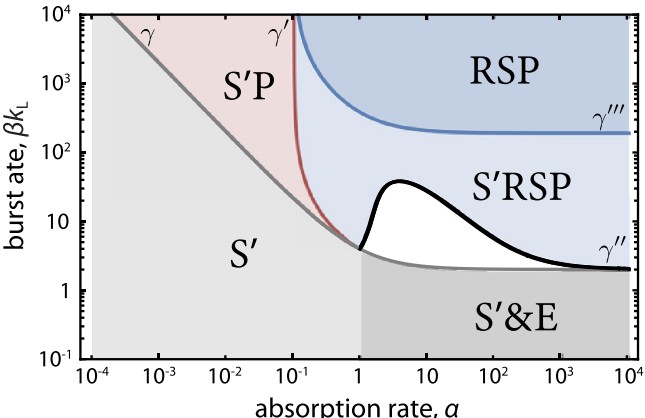

**Fig. 5 Phase diagram of the CRISPR spacer loss model.** The resistant phenotype $R$ generates the loss-of-spacer phenotype $S$ which bears the cost of resistance without the benefit of immunity. $S'$ corresponds to a phage-sensitive phenotype of a strain that lacks CRISPR and does not pay the cost of resistance. The resistant phase contains two regions, $S'$RSP and RSP, separated by the curve $\gamma'''$ across which the system undergoes a transcritical bifurcation that eliminates $S'$ from the population. Coexistence of $S'$ with $RSP$ indicates that $S'$ benefits from the CRISPR system expressed by $R$ without paying the cost. All the rates are in units of per cell division time (1/div). Parameter values are $s = 10^{-3}$, $n_r = 1$, $d' = 1$, $b = 0.9$, $k_L = 1$, $K_m = 0$.

genotypes, corresponding to fixed points of Eq. (3), including a resistance switching strain $R_s$ (phenotype $R$ switches to $S$ at rate $s > 0$), a non-switching resistant strain $R_0$ (phenotype $R$), and a sensitive strain $S$ (phenotype $S$). We consider a single phage type with parameters in the RSP phase for $\alpha > d/n_r$, hence only the $R_s$ strain carries the phage, while the $R_0$ and $S$ strains do not. We note that simple coexistence of $S$ and $P$ is not possible in this part of the phase diagram, as phage with such parameters would drive $S$ to extinction. We analyze dynamics within a patch dominated by one strain when a second strain is introduced initially at low frequency.

For preventative defenses, Fig. 4a shows that $S$ is replaced by $R_s$, as $R_s$ brings phage into the patch together with infected cells that carry and release phage, which infect the $S$ strain and drive it to extinction. However, switching to the sensitive phenotype reduces the growth rate of $R_s$ in the presence of phage, which allows an $R_0$ strain to invade over a timescale $1/s$, and eventually drive both the $R_s$ strain and phage to extinction. Subsequently, an $S$ strain could invade the patch, replacing $R_0$ in the absence of phage. It is therefore crucial to consider how invasion trajectories such as $S \rightarrow R_s \rightarrow R_0 \rightarrow S \dots$, may impact the preservation of resistance at the level of inter-patch dynamics, which we analyze in the next section below.

For immune defenses, the $S \rightarrow R_s \rightarrow R_0$ invasion proceeds in a similar way, but the critical difference is that competitions between $R_0$ and $R_s$ resolve in a surprising manner: the non-switching $R$ phenotype of the $R_0$ strain drives the switching $R$ phenotype of the $R_s$ strain to extinction at rate $s$, but the patch reaches a coexistence of $R_0$, $S$ and $P$, either as a fixed point or limit cycle. The coexistence persists despite the fact that there is no new generation of $S$. Instead, $R_0$ cells act as a phage sink and alleviate the phage pressure on $S$ so that its growth rate matches that of $R_0$, enabling true coexistence.

The stable coexistence of sensitive and immune resistant cells in the presence of phage suggests the possibility that unrelated 'cheater' strains could enjoy the benefit of coexistence with immune strains without paying the cost of resistance. Analysis of the stable phases in that scenario (Fig. 5) indicates that resistance switching can prevent the establishment of such immune defense cheaters. For higher values of $s$, as can be achieved in the mechanism of CRISPR spacer loss[29], higher levels of phage are present, and an $R_s$ strain generates selection pressure against invaders that is proportional to $\beta s$. Invaders whose growth advantage is below that threshold will be driven to extinction (see Methods, 'Model of CRISPR spacer loss').

**Evolutionary stability of the resistance switching strategy**. To analyze the long-term outcomes of competition among a set of

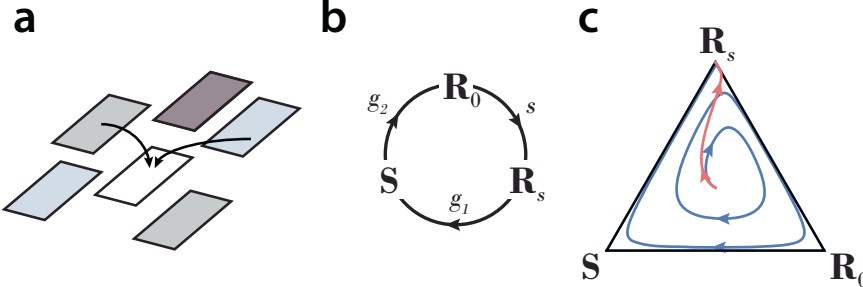

**Fig. 6 Patch invasion dynamics and evolutionary stability.** (**a**) The patch invasion model considers co-invasion events into a cleared patch. Patch clearing events occur at rate $c$. Shading reflects different patch types. (**b**) Illustration of the rock-paper-scissors-type dynamics among a resistance switching strain $R_s$ (which carries phage), a non-switching strain $R_0$, and a sensitive strain $S$. $R_0$ beats $R_s$ with rate $s$, $R_s$ beats $S$ with rate $g_1$, and $S$ beats $R_0$ with rate $g_2$. (**c**) Patch invasion game dynamics approaches a heteroclinic cycle on the boundary for $s > c$ (blue curve) or the evolutionarily stable strategy $R_s$ for $s < c$ (pink curve).

strains across many patches, we apply the ecological model. Each patch is dominated by a single strain, corresponding to a fixed point of the population dynamics equations (3). Patch clearing events occur at rate $c$, and co-invasions of two strains into a cleared patch drive the dynamics of strain frequencies across the ecology, which depend on the set of rates $f_{ij}$ at which strain $i$ outcompetes strain $j$ (Fig. 6a). If a patch is cleared before a competition is resolved, no change in strain frequency will occur. The frequency of strain $i$ across the ecology is denoted by $x_i$, and evolves in time according to the replicator equation of game theory (Eq. (5)), where each strain is a pure strategy in a two-player game with a payoff matrix whose entries depend on $f_{ij}$ and $c$.

We apply the model to the set of strains $i \in \{\mathbf{R}_0, \mathbf{R}_s, \mathbf{S}\}$ in the RSP phase for $\alpha > 1$ (Fig. 6b). For a preventative defense, the competition between $\mathbf{R}_0$ and $\mathbf{R}_s$ will result in the former driving both the latter strain and the phage to extinction, at rate $f_{\mathbf{R}_0\mathbf{R}_s} = s$. In a competition of $\mathbf{R}_s$ versus $\mathbf{S}$, the phage will rapidly drive strain $\mathbf{S}$ to extinction, at rate $f_{\mathbf{R}_s,\mathbf{S}} = g_1$, while competition of $\mathbf{S}$ versus $\mathbf{R}_0$ results in the former outcompeting the latter at rate $f_{\mathbf{SR}_0} = g_2$ due to the cost of defense in the absence of phage (Fig. 4a); we do not require the exact expressions of $g_1$ and $g_2$, and only assume that $g_1, g_2 > c$. The payoff matrix for this game is given by

$$
\begin{array}{c}
\\
\mathbf{R}_s \\
\mathbf{R}_0 \\
\mathbf{S}
\end{array}
\begin{array}{ccc}
\mathbf{R}_s & \mathbf{R}_0 & \mathbf{S} \\
\begin{pmatrix}
1 & 0 & \frac{2g_1}{g_1+c} \\
\frac{2s}{s+c} & 1 & 0 \\
0 & \frac{2g_2}{g_2+c} & 1
\end{pmatrix}
\end{array} . \qquad (6)
$$

For $s > c$ this is a rock-paper-scissors game with no stable Nash equilibria and a heteroclinic cycle on the boundary (Fig. 6c, blue curve)[30]. In this case, the system spends increasingly long times near each of the vertices, while continuing to transition from one to the next indefinitely. It is therefore expected that due to stochastic fluctuations, the ecology will end up in one of the vertices, however which one is generally unpredictable, and could depend on details of the stochastic dynamics and the initial conditions. In contrast, for $s < c$ a strict Nash equilibrium (i.e. an ESS) exists, which corresponds to the resistance switching strain $\mathbf{R}_s$ (Fig. 6c, pink curve). In this regime, the long-term outcome is deterministic, and the $\mathbf{R}_s$ strain is predicted to sweep across the ecology (see Supplementary Note 4).

We can further consider a set of $\mathbf{R}_s$ strains spanning a range of values of $s$, either pre-existing or occurring by mutation (see Methods, 'Game theory of the patch invasion model'). Competitions between strains $\mathbf{R}_s$ and $\mathbf{R}_{s'}$ will resolve in favor of the strain with a smaller switching rate, provided that $|s - s'| > c$. Thus, the distribution of switching rates across the ecology will evolve toward lower values of $s$ until all remaining $\mathbf{R}_s$ strains satisfy $s < c$, rendering $\mathbf{R}_0$ unable to invade. At that point, any remaining $\mathbf{R}_0$ and $\mathbf{S}$ strains will be driven to extinction, and resistance will be preserved thereafter.

We note that the outcomes of patch invasion dynamics depend on the ability to transfer a representative inoculum between patches, which for the $\mathbf{R}_s$ strain includes the transfer of infected cells that release the phage. In practice, inter-patch migration may proceed through bottlenecks which select for large enough switching rates (but still smaller than $c$), so that the infected cells are stably represented in the founder population. A complete description of bottleneck effects and selection is beyond the scope of this work, and will be interesting to study in future work.

It is possible that phage could migrate through other routes, independent of their specific host bacteria. We therefore considered a generalization in which phage constitutes an independent pure strategy $\mathbf{P}$, which corresponds to the host extinction fixed point of Eq. (3). We analyzed the resulting four strategy game in Supplementary Note 5, and showed that $\mathbf{R}_s$ is the unique ESS for $s < c$.

Finally, we comment on the patch invasion dynamics for immune defenses. The $\mathbf{R}_0$ strain can generate $\mathbf{R}_0$-$SP$ coexistence exclusively through co-invasions with the $\mathbf{R}_s$ strain, and can no longer be invaded by the $\mathbf{S}$ strain. The ecology for immune defenses could therefore contain patches with an $\mathbf{R}_0$-$SP$ coexistence, as well as patches occupied by $\mathbf{R}_s$ strains with $s < c$.

## Discussion

By analyzing host-pathogen dynamics within a single patch and modeling bacterial invasions on many patches, we determined the set of conditions in which resistance mechanisms are preserved. We showed that the resistance switching strategy, whereby pathogen-resistant hosts stochastically lose resistance, enables the ecology as a whole to maintain memory of the pathogen. Such ecological memory, an emergent property at the level of ecological dynamics, is the basic requirement for preserving resistance mechanisms over long timescales. Ecological memory has been considered in various contexts[31–33], where it provides a mechanism by which ecosystems robustly adjust to change. In our work, ecological memory corresponds to an evolutionarily stable strategy that maintains a low level of pathogen across the ecology. Our analysis, which bridges from molecular and cellular to population and ecological levels, shows that a non-zero failure rate of defense mechanisms, which reduces host growth rate within a single patch, protects those same mechanisms from eventual loss within the ecology.

Spontaneous loss of a preventative defense was observed experimentally to enable persistence of phage at low frequencies[24,25,34,35]. Similarly, loss of immune defenses is known to occur in CRISPR systems[29,36,37], and modeling has shown that this loss could be responsible for coexistence of phage and bacteria[38], which was experimentally observed in[39]. Our work unifies these observations by providing the critical context of ecological dynamics and memory, and thereby establishes a mathematical basis to analyze evolutionary maintenance of resistance mechanisms. We showed that the same resistance switching strategy that enables ecological memory can maintain multiple phage strains with different combinations of infection rates and burst sizes. Ecological memory can therefore involve a diverse collection of phage types that coexist stably with the host bacterium, which has implications for microbial ecosystem diversity and stability. Further extension of our modeling approach for immune defenses that accounts for CRISPR spacer acquisition and evasion by phage[40–44] may be fruitful in identifying novel dynamics and strategies of generation and maintenance of ecological memory. Additional applications may be relevant in abortive immune systems, where spatial structure has been shown to promote evolution of altruistic resistance mechanisms[45–47].

At the molecular level, the phage-bacteria interaction has long been the subject of biophysical studies[27,28,48,49], including a seminal work on chemoreception[26]. However, there has been a major gap in understanding how the molecular details of the phage-bacteria interaction impact processes at higher levels of organization, in particular at the population and ecological levels. For example, a change in the binding free energy of a phage receptor protein impacts the rate of infection and thus could have a direct fitness consequence which natural selection can act on. Here, by explicitly calculating the form of the infection rate $k_I$ in terms of biophysical parameters (Eq. (2)), we showed that the specific molecular mechanism of resistance has a large effect on host-pathogen dynamics.

In contrast to preventative defenses, immune mechanisms act as phage sinks, absorbing and removing phage from the environment. Phage-sensitive strains, which do not pay the cost of immunity, can act as 'cheaters' by exploiting immune strains to enable their own survival in the presence of phage. On the one hand, this means that immune defenses cannot be eliminated by faster-growing sensitive strains, as these depend on the mutualism for survival. On the other hand, cheating reduces the long-term growth potential of immune strains and may thus select for anti-cheating strategies. We found that resistance switching by immune strains increases the amount of phage in the environment and can be used to select against cheater strains.

The models introduced here enable testing and validation in laboratory experiments[50,51]. Our single patch formulation corresponds to a well-mixed population of bacteria and phage growing in rich or limited media and maintained in a proliferating state by dilution. The host-pathogen interaction term, $k_I(t)$, was constructed by considering phage-receptor binding interactions, yielding a general form applicable across different regimes of host and phage densities, spanning different experimental scenarios. A prediction of our single-patch models is that immune and sensitive strains can coexist stably with phage, which can be directly tested by growing mixtures of bacterial strains with and without a CRISPR system in the presence of phage. Depending on the phage burst and infection rate parameters, our model predicts whether or not coexistence is possible. Our multi-patch ecological model can be tested in multi-well plate format experiments using the $\lambda$-phage system, where each well corresponds to a patch and is inoculated with one of three E. coli strains, $R_0$ (e.g. lamB deletion), S (e.g. constitutive lamB), and $R_s$ (wild type) with phage, in media and grown to saturation. Daily dilution into fresh plates would be performed such that each well receives inocula from two randomly chosen wells of the saturated culture. Ecological dynamics are observed by tracking the prevalence of resistance across the plate.

While our modeling considered phage-bacteria interactions, the general principles that we identified are relevant in other systems, including epidemiological dynamics[52] and host-parasite interactions[53–56]. In particular, our formulation of patch invasion dynamics using game theory, together with the mechanism of ecological memory, may be applicable to the maintenance of pathogen resistance mechanisms in plants, as the costs of such resistance are well-known and patch dynamics models are widely used in plant ecology[3,57,58]. As observed in Ref. [24], resistance switching in bacteria not only sustains the growth of phage but also allows the pathogen eventually to evolve to infect the host through a different pathway. Ecological memory, which allows sustained coexistence of hosts and pathogens, might thus be relevant for future studies of co-evolutionary dynamics in a range of systems.

## Methods

**Phage-receptor binding and infection rate.** In the quasi-steady-state approximation of Eq. (1), $r_b$ is taken to be at steady-state,

$$\frac{dr_b}{dt} = q\, r_f P_f - (\alpha + q') r_b = 0 . \tag{7}$$

By substituting $r_f = r - r_b$ and $P_f = P - r_b$ for free receptor and free phage concentrations, and solving the quadratic equation for $r_b$ we obtain

$$r_b = \frac{K_m + r + P}{2} \left( 1 - \sqrt{1 - \frac{4rP}{(K_m + r + P)^2}} \right), \tag{8}$$

known as the Morrison equation[59], where $K_m = (q' + \alpha)/q$ is the Michaelis constant. We expand in the small parameter $\epsilon \equiv 4rP/(K_m + r + P)^2$ to the first order, noting that we always have $\epsilon < 1$,

$$r_b = \frac{rP}{K_m + r + P} + \mathcal{O}(\epsilon^2) . \tag{9}$$

Using $r = n_r A$ above, we obtain the total phage absorption rate

$$\frac{dP}{dt} = -\alpha r_b = -\frac{\alpha n_r A P}{K_m + n_r A + P} , \tag{10}$$

which yields the per cell infection rate given in Eq. (2).

The value of $K_m$ can be determined from experiments using E. coli and $\lambda$ phage. In the Berg-Purcell limit of a perfectly absorbing host cell ($n_r \to \infty$), phage arrive at the cell surface at rate $k_{max}P$ (Ref. [26]). For finite $n_r$, the rate is given by $k_{max}n_r P/(n_c + n_r)$, where $n_c$ is a constant[26], which yields the per-receptor rate constant $q = k_{max}/(n_c + n_r)$ in Eq. (1). Measured values of $k_{max}$ are on the order of $10^{-11} - 10^{-10}$ cm$^3$ s$^{-1}$, while $q'$ and $\alpha$ are $\sim 10^{-3}$ s$^{-1}$ and $n_c \sim 10^2$ (Ref. [27]). From these, we find $K_m/(n_r + n_c) \sim 10^7 - 10^8$ cm$^{-3}$. This estimate shows that most prior experiments performed with E. coli and $\lambda$ (e.g. Refs. [17,24,25,35]) are consistent with the high binding affinity limit, $n_r A, P \gtrsim K_m$. The low binding affinity limit, $n_r A, P \ll K_m$, yields a Lotka-Volterra interaction term $A \cdot P$ in the population dynamics equations (3), and may be appropriate in other systems. We assume high binding affinity ($K_m \to 0$) in the main text, and analyze the Lotka-Volterra interaction in Supplementary Note 1. We have $\epsilon \ll 1$ for $K_m \gg n_r A, P$, or for $K_m \ll n_r A, P$ if either $P/(n_r A) \ll 1$ or $P/(n_r A) \gg 1$, i.e. in the limits of low or high MOI per receptor. We note that because the relevant MOI is per receptor, most experimental conditions have $P/(n_r A) \ll 1$ and the lowest order term in Eq. (9) is sufficient.

**Population structure and control of total biomass.** To analyze the dynamics of population structure, we compute the relative biomass fractions of each subpopulation while holding the total biomass constant. The total biomass density is given by $B \equiv v_H(R + S + I) + v_P P$, where $v_H$ and $v_P$ are the average mass of a host cell and a phage, respectively, and we define $v \equiv v_P/v_H$. The biomass fractions are given by $f_R = v_H R/B$, $f_S = v_H S/B$, $f_I = v_H I/B$, and $f_P = v_P P/B$, and specify a point in the unit simplex. The host infection rate is expressed as

$$k_I(t) = \alpha n_r \frac{f_P}{K'_m \nu_P + n_r \nu f_A + f_P} , \tag{11}$$

where $K'_m \equiv K_m/B$ and $f_A = v_H A/B$ is the biomass fraction of phage-absorbing hosts. The population dynamics equations on the simplex are given by

$$\begin{aligned}
\dot{f}_R &= (b - s)f_R - \lambda(t)f_R, \\
\dot{f}_S &= df_S + sf_R - k_I(t)f_S - \lambda(t)f_S, \\
\dot{f}_I &= k_I(t)f_S - k_L f_I - \lambda(t)f_I, \\
\dot{f}_P &= -\nu k_I(t)f_A + \nu\beta k_L f_I - \lambda(t)f_P
\end{aligned} \tag{12}$$

where $\lambda(t) = bf_R + df_S - k_L f_I + v\beta_I - vk_I(t)f_A$ enforces the condition $\dot{B} = 0$ in Eq. (3). We analyze the high binding affinity limit ($K_m \to 0$), and show that in this case the stability of the fixed points, which determines the phase structure, does not depend on the choice of $v$. We then determine how inclusion of $K_m > 0$ modifies the stability of fixed points. We note that turbidostat control maintains constant host biomass, i.e. not including phage biomass, and yields equations that can be mapped by a smooth, 1-to-1 mapping to the equations above; the linear stability analysis in a turbidostat therefore matches that obtained at the corresponding fixed points on the simplex (Supplementary Note 1).

**Linear stability analysis.** We perform linear stability analysis of the fixed points of Eq. (12), by computing the eigenvalues of the Jacobian matrix, whose elements are $J_{ij} = d\dot{f}_i/df_j$. Since $\sum_i f_i = 1$, we express $f_P = 1 - f_R - f_S - f_I$ and solve a reduced system consisting of the first three equations in (12).

The host extinction fixed point E corresponds to the solution $(f_R, f_S, f_I) = (0, 0, 0)$. The eigenvalues of the Jacobian evaluated at E are

$$\begin{aligned}
L_1^E &= d - n_r\alpha, \\
L_2^E &= -k_L, \\
L_3^E &= b - s.
\end{aligned} \tag{13}$$

Since $L_3^E > 0$, extinction is always unstable in the presence of resistant phenotype. Analysis of host extinction stability for a system with $f_R = 0$ is presented in Supplementary Note 2.

The S phase fixed point corresponds to the solution $(f_R, f_S, f_I) = (0, 1, 0)$ which exists in the whole phase diagram $\{(\alpha, \beta): \alpha > 0, \beta > 0\}$. Eigenvalues of the Jacobian matrix are

$$\begin{aligned}
L_{1,2}^S &= -d - \frac{\alpha + k_L}{2} \pm \frac{1}{2}\sqrt{(\alpha - k_L)^2 + 4\alpha\beta k_L} \\
L_3^S &= -d + b - s,
\end{aligned} \tag{14}$$

which are negative for $d > b - s$ and $\beta k_L < \gamma$, where

$$\gamma \equiv (d + k_L)\frac{d + \alpha}{\alpha} , \tag{15}$$

hence the S phase is stable in the region $\{(\alpha, \beta): \beta k_L < \gamma\}$.

The SP phase fixed point corresponds to the solution $(f_R, f_S, f_I) = (0, f_S^{SP}, f_I^{SP})$, which exists throughout the region $\{(\alpha, \beta): \alpha < d/n_r, \beta k_L > \gamma\}$ and within part of the region $\{(\alpha, \beta): \alpha > d/n_r, \beta k_L < \gamma\}$. One of the eigenvalues of the Jacobian evaluated at the SP fixed point is

$$L_1^{SP} = b - s - \frac{(d - \alpha n_r)(\beta k_L - d - k_L)}{\beta k_L - (1 + n_r)(d + k_L)}, \quad (16)$$

while the other two eigenvalues are both negative in the region $\{(\alpha, \beta): \alpha < d/n_r, \beta k_L > \gamma\}$. Solving the $L_1^{SP} < 0$ condition in this region yields the location of the stable SP phase as $\{(\alpha, \beta): \alpha < d/n_r, \gamma < \beta k_L < \gamma'\}$, where

$$\gamma' \equiv (d + k_L)\left[1 + \frac{n_r(b - s)}{b - s - d + n_r\alpha}\right]. \quad (17)$$

The RSP phase fixed point corresponds to the solution $(f_R, f_S, f_I) = (f_R^{RSP}, f_S^{RSP}, f_I^{RSP})$ satisfying $f_R^{RSP}, f_S^{RSP}, f_I^{RSP} > 0$, which exists in the region $\{(\alpha, \beta): \beta k_L > \gamma\}$, and its stability is determined by considering eigenvalues that can also be in the complex domain. The RSP phase can therefore contain stable fixed points and periodic dynamics that emerge in the regions where there are no stable fixed points. The existence and size of these regions in the phase diagram will depend on which of the two models of resistance we consider. To find the location of the curve $\gamma''$ where complex-conjugate eigenvalues become purely imaginary, and therefore can lead to a Hopf bifurcation of the dynamical system, we consider the characteristic equation, which is a cubic polynomial in eigenvalues $L$ with real coefficients $a_i$: $P(L) = L^3 + a_2 L^2 + a_1 L + a_0 = 0$. The solutions will satisfy Viète's formulas $L_1 + L_2 + L_3 = -a_2$, $L_1(L_2 + L_3) + L_2 L_3 = a_1$, and $L_1 L_2 L_3 = -a_0$. Let $L_1$ be the real eigenvalue, which is negative when $a_0 > 0$, and let $L_2$ and $L_3$ be the complex conjugate eigenvalues. The fixed point will become unstable when the real parts of $L_2$ and $L_3$ vanish, which when used with Viète's formulas gives a condition $a_1 a_2 = a_0$ that we solve for $\beta k_L$.

**Plotting flow diagrams.** In the analysis above, the specific value of $\nu$ has no effect on the stability of the phases, since the bifurcation curves $\gamma$, $\gamma'$, and $\gamma''$ are determined by the eigenvalues of the Jacobian which were shown to be independent of $\nu$. To plot the flow diagrams in Figs. 2a–d & 3a–b, we set $\nu = 1$, which shows the relative abundances of hosts and phage in a population. Points in the interior of the diagram can be read by following the dotted gridlines to each edge, as illustrated in Supplementary Fig. 1.

**Stability analysis at low binding affinity.** We examined the stability of phases for $K_m > 0$ by considering its modifications to the host infection rate. Since $B$ is held constant under the dynamics in Eq. (12), $K'_m$ is likewise a constant. For immune defenses, where $f_A = f_R + f_S + f_I$, the infection rate for $K'_m > 0$, given in Eq. (11), can be re-written as

$$k_I(t) = \alpha' n'_r \frac{f_P}{n'_r \nu f_A + f_P}, \quad (18)$$

where we introduced rescaled parameters $n'_r = (n_r + K'_m \nu_H)/(1 + K'_m \nu_P)$ and $\alpha' = \alpha n_r/(n_r + K'_m \nu_H)$. Since this expression matches the functional form of the infection rate in the high binding affinity limit, the phase diagram of immune defenses for $K'_m > 0$ maintains the structure shown in the high binding affinity limit, with replacements $\alpha \leftrightarrow \alpha'$ and $n_r \leftrightarrow n'_r$.

For preventative defenses, we obtain

$$k_I(t) = \alpha' n'_r \frac{f_P}{n'_r \nu f_A + f_P + f_R \frac{K'_m \nu_p}{1 + K'_m \nu_p}}. \quad (19)$$

The linear stability analysis of the S and SP phases where $f_R = 0$ recovers the same results as for the immune defenses with $\alpha'$ and $n'_r$, while the RSP phase could potentially include limit cycles.

**Phage decay.** Here we consider the effect of including phage decay at rate $\delta$ in the model, by including a term $-\delta P$ in the expression for $\dot{P}$ in Eq. (3). We analyze the fixed points as above to obtain the formulae for bifurcation curves:

$$\gamma_\delta = (d + k_L)\frac{d + \alpha + \delta}{\alpha} = \gamma\left[1 + \frac{\delta}{\alpha + d}\right]$$

$$\gamma'_\delta = (d + k_L)\left[1 + \frac{n_r(b - s + \delta)}{b - s - d + n_r\alpha}\right] = \gamma'\left[1 + O\left(\frac{\delta}{\alpha + b}\right)\right] \quad (20)$$

Since phage decays typically on the order of days[60], such corrections are small and only slightly shift bifurcation curves.

**Parameter dependence of phase diagrams.** We examined the structure of phase diagrams when model parameters are varied, including $k_L$, $n_r$, $d$, and $b$. Figures 2 and 3 show results for a minimally sensitive phenotype ($n_r = 1$), while increasing the number of receptors per cell moves the phase boundary $\gamma'$ separating SP and RSP phases to the left, expanding the domain of the RSP phase (Supplementary Fig. 7). Decreasing the cost of a defense mechanism given by the growth rate difference $d - b$ likewise moves the $\gamma'$ boundary to the left (Supplementary Fig. 8).

Both dependencies can be seen from the exact expression for $\gamma'$, which has a vertical asymptote at $\alpha = (d - b + s)/n_r$ (Eq. (17)). The phase boundary $\gamma$ separating RSP from S and E phases is independent of $n_r$ and $b$. Phase diagrams were qualitatively unchanged when the phage lysis rate was varied (Supplementary Fig. 9), since $k_L$ affects only the overall scale in the expressions for $\gamma$ and $\gamma'$. We additionally examined the possibility that a phage that injected DNA into a cell remains bound to the receptor and blocks it to further phage absorption (Supplementary Fig. 10). Phage burst sizes vary by orders of magnitude among different phages grown on similar hosts (e.g. $\beta \simeq 10^2$–$10^4$ in *E. coli*[61]) and a tradeoff in burst rate of the form $\beta k_L \approx$ constant[62,63] indicates that changes in $k_L$ may be compensated by changes in $\beta$. Likewise, changing the growth rate $d$ does not impact the phase diagram, as all rates are expressed in units of per cell division time. Phage are known to decay at rates that are several orders of magnitude lower than cell division rates[60] and therefore phage decay does not impact the phase structure (Eq. (20)). To assess the importance of density-dependent growth on the results, we implemented the model using chemostat growth with a limiting resource, which displays a similar phase structure (Supplementary Note 1 & Supplementary Fig. 11).

**Model of CRISPR spacer loss.** We modify the immune defense model to account for a loss-of-spacer phenotype, $S$, which is sensitive to phage but pays the cost of expressing the CRISPR system (i.e. it grows at the same rate as $R$), and which occurs by switching from $R$ at a spacer loss rate $s$. We examine the coexistence of the resistance switching strain with a sensitive strain consisting of phenotype $S'$ which does not express CRISPR and grows at rate $d' > b$. The dynamics are given by

$$\dot{R}(t) = (b - s)R(t) - \lambda(t)R(t),$$
$$\dot{S}(t) = bS(t) + sR(t) - k_I(t)S(t) - \lambda(t)S(t),$$
$$\dot{S'}(t) = d'S'(t) - k_I(t)S'(t) - \lambda(t)S'(t), \quad (21)$$
$$\dot{I}(t) = k_I(t)[S(t) + S'(t)] - k_L I(t) - \lambda(t)I(t),$$
$$\dot{P}(t) = -k_I(t)A(t) + \beta k_L I(t) - \lambda(t)P(t),$$

where $A = R + S + S' + I$. In this system there exist five stable fixed points that determine late-time population structure: the previously described S, SP and E phases, and two new phases that carry resistance, the $S'$RSP phase where all hosts coexist, and the RSP phase in which $S'$ is not present. Figure 5 shows the diagram of these phases separated by curves across which the system undergoes transcritical ($\gamma$, $\gamma'$, $\gamma'''$) and Hopf ($\gamma''$) bifurcations.

Starting from a point in the $S'$RSP phase and increasing phage burst rate, the frequency of $S'$ decreases until it becomes exactly zero at the location of the curve $\gamma'''$. The dynamics transitions to a stable fixed point which has $S' = 0$. Stability of the RSP phase implies that any transient increases of $S'$ frequency, e.g. through random mutations or immigration events, will decay exponentially. The location of this transition is controlled by the fitness difference between $S'$ and $S$ which includes the cost of resistance, and the rate of spacer loss:

$$\gamma''' \equiv \frac{(b + k_L)(d' - b + s)(n_r\alpha - d' + (b - s)(1 + n_r))}{s(n_r\alpha - d' + b - s)} \quad (22)$$

For $\alpha \gg b$ we obtain the lower bound on burst size for which $S'$ is removed:

$$\beta \approx \frac{(b + k_L)(d' - b + s)}{sk_L} \quad (23)$$

Therefore, for $\beta s \gtrsim \Delta f$, where $\Delta f = d' - b + s$ is the fitness difference between $S'$ and $R$, resistance switching suppresses invasions by $S'$. Increasing the switching rate increases the selection against $S'$.

For simplicity, in the above resistance switching strain we considered only spacer loss and did not include CRISPR loss. More generally, both spacer loss and CRISPR loss could occur in a resistance switching strain, further increasing the selection against strains that lack a CRISPR system. There also exist cases in which a CRISPR system exhibits no detectable constitutive costs[64]. In such cases, resistance switching via CRISPR loss corresponds to the above model in which both $R$ and $S$ grow at the same rate, and the above analysis shows that it would select against faster-growing sensitive strains whose growth advantage is unrelated to CRISPR.

**Game theory of patch invasion model.** We analyzed the payoff matrix given in Eq. (6), where the pure strategies are $\mathbf{R}_s$, $\mathbf{R}_0$, and $\mathbf{S}$, given $g_1, g_2 > c$. Each pure strategy is a fixed point of the replicator equation (Eq. (5)), and for $s > c$ no pure strategy Nash equilibrium exists. For $s \le c$, $\mathbf{R}_s$ is a Nash equilibrium, and for $s < c$ it is a strict Nash equilibrium, hence an ESS. For $s = c$, $\mathbf{R}_0$ is an alternative best reply to $\mathbf{R}_s$, and since $\mathbf{R}_s$ does not beat $\mathbf{R}_0$, $\mathbf{R}_s$ is not an ESS. Depending on parameter values, two other fixed points can exist, one on the $\mathbf{R}_s$-$\mathbf{R}_0$ edge for $s < c$ and the other in the simplex interior, yet neither one of these can be stable. See Supplementary Note 4 for derivation, Supplementary Note 5 for the case of independently dispersing phage, and Supplementary Fig. 12 for representative phase plots.

Next we considered a model with two switching strains, $\mathbf{R}_s$ and $\mathbf{R}_{s'}$, with $s - s' > c$. In a similar fashion, we obtain the payoff matrix $\phi$:

$$\begin{array}{c}\\ \mathbf{R}_s \\ \mathbf{R}_{s'} \\ \mathbf{R}_0 \\ \mathbf{S}\end{array}\begin{array}{cccc}\mathbf{R}_s & \mathbf{R}_{s'} & \mathbf{R}_0 & \mathbf{S} \\ \left( \begin{array}{cccc} 1 & 0 & 0 & \frac{2g_1}{g_1+c} \\ \frac{2(s-s')}{s-s'+c} & 1 & 0 & \frac{2g_1'}{g_1'+c} \\ \frac{2s}{s+c} & \frac{2s'}{s'+c} & 1 & 0 \\ 0 & 0 & \frac{2g_2}{g_2+c} & 1 \end{array} \right)\end{array} \quad (24)$$

The stable fixed point corresponds to a pure $\mathbf{R}_{s'}$ strategy for $s' < c$. If both $s$ and $s'$ are larger than $c$, there is no interior equilibrium and the system will approach the boundary of the simplex $\sum_i x_i = 1$ whose faces correspond to a reduction of the game theory model to a subset of three strains. The unstable fixed point on the $\{\mathbf{R}_s, \mathbf{R}_0, \mathbf{S}\}$ face repels interior orbits and the system transitions to a heteroclinic cycle on the $\{\mathbf{R}_{s'}, \mathbf{R}_0, \mathbf{S}\}$ face. Equivalently, the strain with the highest switching rate is driven to extinction as it gets displaced by the strain with lower switching rate and the ecology reduces to the $\{\mathbf{R}_{s'}, \mathbf{R}_0, \mathbf{S}\}$ patch invasion game.

We generalized to a large number of switching strains with $s > s' > s'' > \ldots > c$ where the difference between each pair of switching rates is greater than $c$. The invasion diagram for three switching strains is:

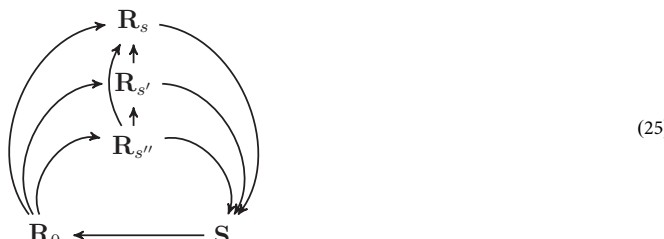

$$(25)$$

If all the strains have switching rates above $c$, the ecology will undergo a similar reduction where the strains with highest switching rates progressively go extinct until one of the rates evolves to become smaller than $c$, at which point it becomes an ESS.

## Data availability
No datasets were generated or analyzed during the current study.

## Code availability
Code used to produce figures is available at github.com/kussell-lab/ecological-memory.

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

## Acknowledgements

This work was supported by NIH grants R01GM097356 and R01GM120231.

## Author contributions

A.S. and E.K. performed the research and wrote the manuscript.

## Competing interests

The authors declare no competing interests.
