## [Peer Review File · Nature Communications]

REVIEWER COMMENTS

Reviewer #1 (Remarks to the Author):

Skandata and Kussell build a clear and thorough mathematical model to explain how bacterial defense mechanisms can be evolutionary stable in the face of cheaters. Their hypothesis – a “resistance switching strategy” – is that the cheaters allow the maintenance of pathogens, which in turn provides a fitness benefit to having a defense mechanism. Essentially, this argues for a form of group selection that preserves the “ecological memory” of a pathogen. The introduction and discussion do a nice job tying the mathematical core of the paper to its larger biological relevance. Their model provides a formal, quantitative resolution to a vexing problem in microbial ecology and suggests experimental tests that could be used to validate their predictions.

I reviewed a previous version of the paper submitted to another journal, and the authors already did a nice job addressing my comments with regard to modeling choices – so I have nothing specific to add here aside from a minor text suggestion at the bottom of pg 2:

In the following clause: “while in strains that do not switch off”, it is jarring / unclear which strains are being referred to. The previous sentences talk generally about E coli and λ and imply that all strains undergo mutations potentially leading to renewed susceptibility. Similarly, the last sentence in this paragraph (at the start of pg 3) says that large population sizes ENSURE sensitive mutants, which again implies that all strains undergo such potential mutations. Please clarify this paragraph.

Reviewer #2 (Remarks to the Author):

This study claims that imperfect resistance can be evolutionary stable because it maintains the coexistence with the pathogen and prevents the ultimate loss of the defense. The analysis of multiple scenarios is interesting but sometimes confusing. Also, I am surprised the authors do not discuss the possibility that imperfect resistance may only be a developmental constraint (it is simply impossible to generate R_0 hosts) which would be a more parsimonious explanation than adaptive evolution.

This theoretical manuscript is difficult to read. Many different scenarios are discussed (different types of resistance, single or metapopulation framework) which blurs the message.

If we focus on the single population scenario then it is possible to see the evolution of R_0 hosts (with or without coexistence with S and pathogens) which suggests that the evolution of R_s is not expected (in contrast with the claim of the authors but I guess this single population is just one step of their argument).

But in the metapopulation model I don't understand the assumption regarding the phage dynamics. It seems that it is impossible to get rid of the pathogen. The authors assume phage is always present with R_s but why is it not possible to see phages in S demes? I am really confused about this model. I like the idea that spatial structure could affect this evolutionary dynamics but the ecology is very unclear and I am not convinced by the present model.

There is also an issue regarding the lack of mutation in the deterministic model. There is a mutation process between R and S and it is unclear to me how it would interact with a very low value of s .

I am also unclear about the deterministic nature of the model. The authors discuss the implications of R_0 evolution on the extinction of the phage (and the reintroduction of the phage later on). Also, the authors notice several times that very low strain frequencies may lead to extinction. Hence stochasticity is likely to have a huge effect on the ultimate evolutionary outcome and these stochastic events are not formally described and modelled.

Another thing I am unclear with is the effect of density dependence on the evolutionary dynamics. There is no density dependence in bacterial growth rate in (1). In M2 you incorporate density dependence but why do you assume bacteria and phage are subject to the same density dependence? Bacteria should compete against each other, not against phage. And phages do not need density dependence if there is density dependence in the bacteria.

Reviewer #3 (Remarks to the Author):

I have assisted (as a trainee at a different career stage) in the review of this manuscript at another journal. I felt quite positively about the work then, and believed it should have been offered the opportunity for revision at the time (the issues that came up, in my mind, were primarily aesthetic).

This manuscript presents a careful and incredibly thorough analysis of an important and unsolved question in the evolution of virus-host interactions – if antiviral defense systems are so efficient (as seen in laboratory experiments), why do they still exist? (i.e., why haven't they driven phage extinct locally). The authors take classical models of phage-host interactions, generalize them, and then place them in a metapopulation framework. In doing so they, as they state, examine a virus-host

system at scales spanning the molecular to landscape level. Their final conclusion, that the loss of immunity can paradoxically lead to its evolutionary maintenance, is compelling and builds on other recent theoretical work on immune loss and “leaky” resistance.

I have a few larger comments that I think can mostly be addressed via relatively minor changes to the text:

- The “costs” of defense, specifically with respect to CRISPR systems, are referenced throughout the manuscript, but these costs aren’t always actually observed in experimental systems. For example, in the *Pseudomonas aeruginosa* model system virus-resistant surface mutants can appear that have no measurable growth cost in culture. Similarly, in this system CRISPR has no constitutive cost in culture as shown with competition experiments (<https://doi.org/10.1016/j.cub.2015.01.065>, <https://doi.org/10.1038/s41586-019-1662-9>). The cost of CRISPR is only felt during viral infection, and this is likely a byproduct of transient expression of viral genes rather than an actual cost of the system itself (<https://doi.org/10.1038/s41396-020-00794-w>). Defense doesn’t always have to be costly, though certainly it can be. I don’t necessarily think this means that the model needs to be changed, but this fact should be recognized in the manuscript and some attention should be paid to the case where $b=d$. Special note of the sometimes-cost-free nature of CRISPR should be made in the CRISPR modeling section. Autoimmunity is cited as a primary cost of CRISPR immunity, but CRISPR systems have a number of ways of recognizing self versus non-self that mean rates of self-targeting are likely to be very low (<https://doi.org/10.1038/nature14302>, <https://doi.org/10.1016/j.tim.2020.02.005>).

- I am a little confused by the co-invasion analysis in the last paragraph of page 8. It seems to me that the authors assume that both strains arrive at the patch at the same time, and that competition can continue until the next clearance event. It seems much more likely to me that strains are likely to arrive one after the other, and that competition will only be able to continue until establishment of one strain (τ_{est}), where presumably $\tau_{est} \ll c$? I doubt this will change the overall result, but I imagine it would slow down the rate at which the system approaches Nash equilibrium and also the threshold s values that permits this equilibrium? I may be missing something here – but in any case some expansion of this section is necessary as it is an essential piece of analysis to the main result and will be largely opaque to less mathematically inclined readers.

- My interpretation of establishment is that resident host have used all available resources in a patch. Thus, clearance of these hosts from a patch wouldn’t necessarily provide new resources for the next invader to grow. I wonder if patch creation is a better analogy (e.g., marine snow, resource particles being created in various environments). I don’t think this would require any modifications to the model, but c would be interpreted as the patch creation rate? Maybe something to at least mention – since the model seems general enough to capture both possibilities.

- I wonder what the authors expect would happen if phage are able to disperse independently of their hosts (this is to be expected in many natural systems where phage particle residency time in the environment is long). Again, I'm not sure this needs to be explicitly modeled but it would be nice to see a line or two discussing this possibility or why it can be excluded.

- I am not convinced that the inclusion of reversible phage binding contributes in any way to the main conclusions of the manuscript. While this is a nice theoretical exploration, it is unclear why the authors prefer it over the more traditional use of the "Lotka-Volterra" approximation they make note of that has been used in the phage-host modeling field for decades. I think much of the discussion of binding kinetics could be moved to a supplement. If the authors could find specific examples of parameter estimates for phage-host systems where the LV approximation is insufficient I would be more open to the inclusion of these analyses in the main text. I think, out of consideration for less mathematically-inclined readers coming from microbiology, the authors should try to minimize the number of mathematical explorations in the main text that are not absolutely critical to their primary conclusions about immune/resistance loss.

Additionally, I have a number of smaller comments that I think would help improve the readability of the manuscript:

- Please include line numbers in any submitted revision (this makes the reviewing much easier)

- The use of kappa, lower case k, upper case K, and various subscripted k's to all denote different quantities in the main model presentation and section M.1 makes these sections extremely hard to follow. It took me a very long time staring at these equations to realize that lower-case k and kappa are actually different symbols (and are actually inversely related) – I was wondering why a high rate of binding (lower-case k) meant a low binding affinity (kappa) until I realized these were different symbols.

- I don't think the second paragraph of the introduction is necessary and I think it breaks up the logical flow of this section (the previous version of the manuscript without this paragraph read much better)

- The information in the last paragraph on page 4 specifying that the turbidostat model is analyzed in the main text should be moved to the top of the model presentation section – I understand that the authors want to present the most general model first but for readers coming from biology it will be much easier to understand a model tied to a concrete biological system.

- On page 7, the sentence “We consider a single phage type strains do not” needs clarification. It will not be immediately apparent to all readers why this is true, but it is an important point for downstream analysis.

- First paragraph of Discussion, shouldn't “pathogen” be pluralized throughout this paragraph?

- Discussion paragraph 1&2, I'm not convinced how useful “memory” is as an analogy here (what conceptual work is this doing?). I think a much clearer title for example, would be “The loss of phage resistance mechanisms paradoxically leads to their long-term preservation in bacteria”, or something along those lines

- The first part of the 3rd paragraph of the discussion outlining modeling work on “leaky resistance” and immune loss could be moved to the intro – noting that the present manuscript greatly expands on previous theoretical explorations of the loss of antiviral defense.

- The conceptual leap to “cheating” in the last paragraph of page 10/first paragraph of page 11 is abrupt and will not make much sense to readers less familiar w/ evolutionary game theory

- Fig 5B – maybe add strain labels instead of just colors to really emphasize

- Fig S3 – specify growth rate of what on y-axis

REVIEWER COMMENTS

Reviewer #1 (Remarks to the Author):

Skandata and Kussell build a clear and thorough mathematical model to explain how bacterial defense mechanisms can be evolutionary stable in the face of cheaters. Their hypothesis – a “resistance switching strategy” – is that the cheaters allow the maintenance of pathogens, which in turn provides a fitness benefit to having a defense mechanism. Essentially, this argues for a form of group selection that preserves the “ecological memory” of a pathogen. The introduction and discussion do a nice job tying the mathematical core of the paper to its larger biological relevance. Their model provides a formal, quantitative resolution to a vexing problem in microbial ecology and suggests experimental tests that could be used to validate their predictions.

I reviewed a previous version of the paper submitted to another journal, and the authors already did a nice job addressing my comments with regard to modeling choices – so I have nothing specific to add here aside from a minor text suggestion at the bottom of pg 2:

In the following clause: “while in strains that do not switch off”, it is jarring / unclear which strains are being referred to. The previous sentences talk generally about E coli and λ and imply that all strains undergo mutations potentially leading to renewed susceptibility. Similarly, the last sentence in this paragraph (at the start of pg 3) says that large population sizes ENSURE sensitive mutants, which again implies that all strains undergo such potential mutations. Please clarify this paragraph.

We revised this paragraph to clarify this point (lines 27 - 29).

Reviewer #2 (Remarks to the Author):

This study claims that imperfect resistance can be evolutionary stable because it maintains the coexistence with the pathogen and prevents the ultimate loss of the defense. The analysis of multiple scenarios is interesting but sometimes confusing.

We have aimed to improve the clarity of the manuscript throughout.

Also, I am surprised the authors do not discuss the possibility that imperfect resistance may only be a developmental constraint (it is simply impossible to generate R_0 hosts) which would be a more parsimonious explanation than adaptive evolution.

In any given example of pathogen resistance, it is important to consider the possibility that stochastic failure of resistance may be (a) a consequence of biological constraints or (b) an evolved trait. In either case, when there is a cost to resistance the question of how resistance is maintained is relevant, and our theory is applicable.

The best studied example in bacteria – the *E. coli* lambda phage experiments - appears to belong to case (b). First, in this system R_0 strains routinely occur in experiments, and typically correspond to *lamB* mutations (Meyer et al. 2012; Chaudhry et al. 2018; Meyer & Lenski 2020). Second, there is substantial evidence indicating that the *malT* gene contains a hotspot for indel mutations, such that *malT* mutants (*R* phenotype) can stochastically revert (*S* phenotype) at rates that are much higher than baseline mutation rates (Chaudhry et al. 2018; Meyer & Lenski 2020).

We have clarified the presentation of this point in the introduction. We highlight the experimental observation of *lamB* mutants that do not revert, and of *malT* mutations which involve insertions and deletions that do revert (lines 23 - 27). We state that phage are observed to go extinct in the *lamB* mutants, and point out the implication regarding maintenance of resistance (lines 27 - 31).

This theoretical manuscript is difficult to read. Many different scenarios are discussed (different types of resistance, single or metapopulation framework) which blurs the message.

We have sought to improve the readability in our revision.

We have added a new paragraph in the introduction (lines 32 - 49) which presents a kind of roadmap of our multi-level modeling approach. This now allows the reader to see how we will move up from molecular to population to ecological levels, and how each successive level is constructed from the previous one. We have also revised the final paragraph of the introduction (lines 50-57) to clearly state the main message of the paper, and then highlight its secondary implications.

In our presentation of the ecological modeling, we have shifted the presentation first to describe the overall framework (lines 107-153), and then apply it to each of the different types of resistance (lines 266 - 312). This separation of framework and applications should further enhance readability, and strengthen the overall message.

If we focus on the single population scenario then it is possible to see the evolution of R_0 hosts (with or without coexistence with S and pathogens) which suggests that the evolution of R_s is not expected (in contrast with the claim of the authors but I guess this single population is just one step of their argument).

In the single population model for immune defense mechanisms (Fig. 4B), R_0 hosts outcompete R_s (for $s > 0$), and thereafter coexist with the S phenotype in the presence of phage. The same is not true of preventative defense mechanisms (Fig. 4A) where the dynamics involves successive substitutions of R_s , R_0 , S , R_s , etc.

The role of the single population modeling is to identify the fixed points of the dynamics, which are then used to define the relevant deme structures in the metapopulation model (see next comment).

But in the metapopulation model I don't understand the assumption regarding the phage dynamics. It seems that it is impossible to get rid of the pathogen. The authors assume phage is always present with R_s but why is it not possible to see phages in S demes? I am really confused about this model. I like the idea that spatial structure could affect this evolutionary dynamics but the ecology is very unclear and I am not convinced by the present model.

To address the reviewer's specific question about phage dynamics in the model, we added a section in the supplement in which the ecological model is expanded to allow P (phage) to migrate independently of host cells; in this model S and P can co-invade and compete in an empty patch. The analysis of this model yields the same result as before, i.e. R_s is the unique ESS for $s < c$. We refer to this section on lines 304 - 308.

To address the reviewer's more general point about how the ecological model is constructed, we note that the demes that we consider in the metapopulation model are fixed points of the single population dynamics. These fixed points may be stable or unstable. This implies that a deme maintains its internal population structure as long as it is not perturbed by migration. The metapopulation model is introduced specifically to understand how perturbations - i.e. migration events - impact the frequencies of different types of demes.

An SP deme corresponds to a deme that has both S cells and phage. In Figs. 2 & 3, in the blue region for $\alpha > 1$, SP is not a fixed point because the phage parameters are such that lysis dominates over growth and any state containing both S and P would instantaneously change, decreasing the number of S cells. In contrast, S and RSP are fixed points in this region, the latter being a fixed point for any value of the switching rate $s > 0$, while additionally R is a fixed

point for $s = 0$. Thus, when we construct the metapopulation model in this region of the phase diagram (see lines 276-277), we analyze competitions between S , R_s , and R_0 . The same holds in the white region (where a limit cycle exists), except that RSP is not a fixed point here but a periodic orbit.

SP demes are relevant in the pink regions (Figs. 2 & 3) where SP is a fixed point while RSP is not. This region corresponds to a phage with a low enough infection rate such that S outcompetes R even in the presence of phage. In this case, the metapopulation model consists of SP and S demes (as well as R demes if $s = 0$), and since SP strictly dominates in co-invasions, SP demes are the unique ESS.

We have revised the text accordingly to emphasize that the deme structure is built up out of fixed points of the single population model (lines 45-47, 65-66, 108-113, 236, 269), and explain on lines 239-241 that SP is not a fixed point in the RSP phase for $\alpha > 1$.

We thank the reviewer for these comments which helped us improve the presentation.

There is also an issue regarding the lack of mutation in the deterministic model. There is a mutation process between R and S and it is unclear to me how it would interact with a very low value of s .

We have added text discussing this point (lines 192-195). The model that we use does not distinguish between genetic mutations and epigenetic processes as the mechanisms underlying resistance switching. Therefore, if both mutations and epigenetic switching are present, these would both contribute to the effective value of s in the model.

I am also unclear about the deterministic nature of the model. The authors discuss the implications of R_0 evolution on the extinction of the phage (and the reintroduction of the phage later on). Also, the authors notice several times that very low strain frequencies may lead to extinction. Hence stochasticity is likely to have a huge effect on the ultimate evolutionary outcome and these stochastic events are not formally described and modelled.

We have added further discussion of this point at lines 284 - 291.

While we analyze deterministic models, our analysis has implications for both stochastic and deterministic dynamics. In particular, we identify the conditions in which stochastic dynamics will be crucially important to the evolutionary dynamics. This occurs in the case when a heteroclinic cycle exists, for values of $s > c$. In such cases, the outcome is entirely subject to stochastic effects, and the details of that dynamical process become important. Further analysis of that regime could be performed in future work. Our main aim is to point out when resistance switching is evolutionarily stable, and in that regime stochastic effects do not affect the outcome.

Another thing I am unclear with is the effect of density dependence on the evolutionary dynamics. There is no density dependence in bacterial growth rate in (1). In M2 you incorporate

density dependence but why do you assume bacteria and phage are subject to the same density dependence? Bacteria should compete against each other, not against phage. And phages do not need density dependence if there is density dependence in the bacteria.

Density-dependent growth of bacteria is analyzed in Sec. S10.2 in the supplement, in which the chemostat model is described. In that model, the bacteria exhibit density-dependent growth (through the nutrient concentration), while phage are (i) absorbed by bacteria, (ii) produced by lysis of infected cells, and (iii) diluted by medium flow (see Eq. (S3)). We show that a similar phase diagram is obtained in the chemostat model compared to the turbidostat model (Sec. S10), and refer to this result in the main text (lines 230-233).

The model in Sec. M2 (Eq. (10)) is identical to the model given in Eq. (3) (previously Eq. (1)), and corresponds to feedback dilution on total biomass, which we show is equivalent to turbidostat control of host biomass (Sec. S10.1). In this case bacterial growth rates are not density dependent (i.e. b and d are fixed constants). These equations correspond to frequency-dependent growth of different biomass components; this is the simplest possible case of frequency dependence, as it results purely from the constraint that the total biomass is maintained constant (see line 419). Eq. 3 exhibits competition between bacteria, which occurs when one component (e.g. R or S) increases relative to another. Phage and bacteria both contribute to biomass, but each does so with its own biomass coefficient; we show that the stability of fixed points is independent of the values of these coefficients (lines 420-421, 466-468).

We hope our revised presentation of the main model equations has clarified these points.

Reviewer #3 (Remarks to the Author):

I have assisted (as a trainee at a different career stage) in the review of this manuscript at another journal. I felt quite positively about the work then, and believed it should have been offered the opportunity for revision at the time (the issues that came up, in my mind, were primarily aesthetic).

This manuscript presents a careful and incredibly thorough analysis of an important and unsolved question in the evolution of virus-host interactions – if antiviral defense systems are so efficient (as seen in laboratory experiments), why do they still exist? (i.e, why haven't they driven phage extinct locally). The authors take classical models of phage-host interactions, generalize them, and then place them in a metapopulation framework. In doing so they, as they state, examine a virus-host system at scales spanning the molecular to landscape level. Their final conclusion, that the loss of immunity can paradoxically lead to its evolutionary maintenance, is compelling and builds on other recent theoretical work on immune loss and “leaky” resistance.

I have a few larger comments that I think can mostly be addressed via relatively minor changes to the text:

- The “costs” of defense, specifically with respect to CRISPR systems, are referenced throughout the manuscript, but these costs aren't always actually observed in experimental systems. For example, in the *Pseudomonas aeruginosa* model system virus-resistant surface mutants can appear that have no measurable growth cost in culture. Similarly, in this system CRISPR has no constitutive cost in culture as shown with competition experiments (<https://doi.org/10.1016/j.cub.2015.01.065>, <https://doi.org/10.1038/s41586-019-1662-9>). The cost of CRISPR is only felt during viral infection, and this is likely a byproduct of transient expression of viral genes rather than an actual cost of the system itself (<https://doi.org/10.1038/s41396-020-00794-w>). Defense doesn't always have to be costly, though certainly it can be. I don't necessarily think this means that the model needs to be changed, but this fact should be recognized in the manuscript and some attention should be paid to the case where $b=d$. Special note of the sometimes-cost-free nature of CRISPR should be made in the CRISPR modeling section. Autoimmunity is cited as a primary cost of CRISPR immunity, but CRISPR systems have a number of ways of recognizing self versus non-self that mean rates of self-targeting are likely to be very low (<https://doi.org/10.1038/nature14302>, <https://doi.org/10.1016/j.tim.2020.02.005>).

We have modified the statement on lines 17-18, removing the previous more general statement about costs, and added the above reference to Westra et al. (2015) on lines 515-521 of the CRISPR modeling section to point out the possibility of cost-free CRISPR expression. Our model of spacer loss in fact assumes $b = d$, as both *R* and *S* phenotypes have the same growth rate *b*, and is therefore also appropriate for modeling cases of cost-free CRISPR expression, which we now point out.

- I am a little confused by the co-invasion analysis in the last paragraph of page 8. It seems to me that the authors assume that both strains arrive at the patch at the same time, and that competition can continue until the next clearance event. It seems much more likely to me that strains are likely to arrive one after the other, and that competition will only be able to continue until establishment of one strain (τ_{est}), where presumably $\tau_{est} \ll c$? I doubt this will change the overall result, but I imagine it would slow down the rate at which the system approaches Nash equilibrium and also the threshold s values that permits this equilibrium? I may be missing something here – but in any case some expansion of this section is necessary as it is an essential piece of analysis to the main result and will be largely opaque to less mathematically inclined readers.

We have clarified this point in the text (lines 133-135). We do not assume simultaneous arrival. The second strain must arrive within τ_{est} of the first, where $\tau_{est} \ll 1/m \ll 1/c$. Competition continues until either one strain effectively sweeps against the other (which depends on the relative fitness of the strains) or the patch is cleared.

- My interpretation of establishment is that resident host have used all available resources in a patch. Thus, clearance of these hosts from a patch wouldn't necessarily provide new resources for the next invader to grow. I wonder if patch creation is a better analogy (e.g., marine snow, resource particles being created in various environments). I don't think this would require any modifications to the model, but c would be interpreted as the patch creation rate? Maybe something to at least mention – since the model seems general enough to capture both possibilities.

We have edited the text to address this point (line 117).

- I wonder what the authors expect would happen if phage are able to disperse independently of their hosts (this is to be expected in many natural systems where phage particle residency time in the environment is long). Again, I'm not sure this needs to be explicitly modeled but it would be nice to see a line or two discussing this possibility or why it can be excluded.

We added this analysis in the supplement (Section S12). We show that independently dispersing phage can be modeled as a separate strategy, and that the ESS condition remains unchanged. We added a comment about this in the main text (lines 304-308).

- I am not convinced that the inclusion of reversible phage binding contributes in any way to the main conclusions of the manuscript. While this is a nice theoretical exploration, it is unclear why the authors prefer it over the more traditional use of the "Lotka-Volterra" approximation they make note of that has been used in the phage-host modeling field for decades. I think much of the discussion of binding kinetics could be moved to a supplement. If the authors could find specific examples of parameter estimates for phage-host systems where the LV approximation is insufficient I would be more open to the inclusion of these analyses in the main text.

We have substantially streamlined the receptor binding kinetics section, and added parameter measurements for the E.coli/Lambda system which demonstrate that the Lotka-Volterra approximation does not hold at the typical bacterial densities found in previous experiments on this system (lines 394-407).

We believe the molecular level modeling is critical as it provides the connection between the level at which mutations act (e.g. modifying receptor affinity, number of receptors, absorption rates, etc.) and their impact on population dynamics. The derived functional form of the infection rate is a consequence of these interactions. Without such modeling one cannot determine when the Lotka-Volterra approximation holds.

Because Lotka-Volterra models are common in the field, we include the Lotka-Volterra analysis in the supplement (Fig. S10). Our analysis expresses the pre-factor of the Lotka-Volterra interaction in terms of molecular parameters, which is useful for studying systems in the dilute limit (lines 83-85).

I think, out of consideration for less mathematically-inclined readers coming from microbiology, the authors should try to minimize the number of mathematical explorations in the main text that are not absolutely critical to their primary conclusions about immune/resistance loss.

We have moved two sections of the methods into supplementary information (now Sec. S4 and S10 in Supplementary Text), and reduced the number of equations in the main text.

Additionally, I have a number of smaller comments that I think would help improve the readability of the manuscript:

- Please include line numbers in any submitted revision (this makes the reviewing much easier)

We added line numbers to the manuscript.

- The use of kappa, lower case k, upper case K, and various subscripted k's to all denote different quantities in the main model presentation and section M.1 makes these sections extremely hard to follow. It took me a very long time staring at these equations to realize that lower-case k and kappa are actually different symbols (and are actually inversely related) – I was wondering why a high rate of binding (lower-case k) meant a low binding affinity (kappa) until I realized these were different symbols.

We have substantially simplified the notation in the Models section (lines 68 - 88), and Methods section M.1 (lines 383 - 393). We removed kappa altogether by using concentration units everywhere, changed the single receptor binding/unbinding rates to q and q' , and denote by K_m the binding constant which is standard notation.

- I don't think the second paragraph of the introduction is necessary and I think it breaks up the logical flow of this section (the previous version of the manuscript without this paragraph read much better)

We edited and moved that paragraph to the discussion (lines 339-347).

- The information in the last paragraph on page 4 specifying that the turbidostat model is analyzed in the main text should be moved to the top of the model presentation section – I understand that the authors want to present the most general model first but for readers coming from biology it will be much easier to understand a model tied to a concrete biological system.

We incorporated the dilution rate directly into the main text equations (Eq. (3)), and added a note about the turbidostat model in the explanation of those equations at lines 101 - 105.

- On page 7, the sentence “We consider a single phage type strains do not” needs clarification. It will not be immediately apparent to all readers why this is true, but it is an important point for downstream analysis.

We added an explanation on lines 239-241.

- First paragraph of Discussion, shouldn't “pathogen” be pluralized throughout this paragraph?

We removed the first three sentences of the discussion, which were redundant with the Introduction, and the word pathogen is now correctly pluralized or singular as necessary.

- Discussion paragraph 1&2, I'm not convinced how useful “memory” is as an analogy here (what conceptual work is this doing?). I think a much clearer title for example, would be “The loss of phage resistance mechanisms paradoxically leads to their long-term preservation in bacteria”, or something along those lines

We use the term “ecological memory” here to highlight the fact that both pathogen and resistance can only be preserved in the ecological context. We believe keeping this term in the title and the discussion is useful as it distinguishes pressures at the single population level, where pathogens and resistance can be lost, from the ecological level where pathogen and resistance are maintained.

- The first part of the 3rd paragraph of the discussion outlining modeling work on “leaky resistance” and immune loss could be moved to the intro – noting that the present manuscript greatly expands on previous theoretical explorations of the loss of antiviral defense.

We find that these points (currently on lines 326-331) are best addressed in the discussion rather than the introduction.

- The conceptual leap to “cheating” in the last paragraph of page 10/first paragraph of page 11 is abrupt and will not make much sense to readers less familiar w/ evolutionary game theory

We added a clarification on line 350.

- Fig 5B – maybe add strain labels instead of just colors to really emphasize

We swapped panels A and B in this figure (currently Fig. 6) for consistency with the order of discussion. We decided not to modify panel A because we reference this figure in order to explain patch dynamics before the strain labels have been defined in the text.

- Fig S3 – specify growth rate of what on y-axis

We edited the figure caption to clarify this is the growth rate of host cells.

REVIEWERS' COMMENTS

Reviewer #2 (Remarks to the Author):

The revised version improved the clarity but I think the paper remains difficult to read.

The main result is illustrated in Figure 6c (pink line) when $s < c$. It shows that in a deterministic model Rs can invade and outcompete other strains. This shows that spatial structure can favour leaky resistance. The problem is that by the time the reader reaches this result, it has to go through the analysis of multiple scenarios. Also, the dynamics of the metapopulation is still not very clear to me. The sentence: "Patches can exist in two states, saturated or empty" is misleading. There is an establishment time where populations are in an intermediate situation. Besides when you write "Saturated patches are occupied by hosts that dominate the local resources, and cannot be invaded" remains unclear. Phage and bacteria are going extinct and recolonise at the same rate? Why? I know you relax this in the Sup Info but you should discuss this sooner.

If the main result is to say that spatial structure promotes the evolution of altruistic resistance you should discuss earlier models about this (e.g. Débarre et al 2012 Am Nat 179(1) 52-63, Fukuyo et al 2012 Sci Rep 2, 238, Berngruber et al 2013 Ecol Letters, 16: 446-453). Discussing these earlier models would help see what is driving the evolution of leaky resistance. In particular no need to introduce the notion of "ecological memory" (what does it mean?) to explain the effect of spatial structure.

Reviewer #3 (Remarks to the Author):

The authors have adequately addressed my concerns. The addition of S12 Text describing a model with independently dispersing phage is a nice addition. I think this careful and comprehensive theoretical work makes an important contribution to the phage-host literature.

We have addressed all of the editorial requests, and provide detailed responses to the requests in the attached document.

Reviewer Comments

Reviewer #2 (Remarks to the Author):

The revised version improved the clarity but I think the paper remains difficult to read. The main result is illustrated in Figure 6c (pink line) when $s < c$. It shows that in a deterministic model Rs can invade and outcompete other strains. This shows that spatial structure can favour leaky resistance. The problem is that by the time the reader reaches this result, it has to go through the analysis of multiple scenarios.

We are glad that the revision improved the overall clarity of the paper. The remaining concern of the reviewer is related to the analysis of multiple scenarios before the main result in Fig. 6c can be stated. To further improve readability, we have moved the section on parameter dependence from Results to Methods, which shortens the presentation and gets the reader more rapidly to the ecological level.

We note that if we start immediately by writing down the dynamics at the ecological level, the work would be open to a critique that interactions at other levels of organizations (e.g. cellular or single population) might be sufficient to explain the maintenance of resistance. For this reason, it is critical to carefully present and analyze the multilevel model step-wise, and build up understanding at each level. Furthermore, our model yields additional key results, such as the different dynamics of preventative versus immune defenses, the ability of cheater strains to avoid immune defense costs, and other related points, before reaching the ecological level, which will be of interest for the broader readership.

Also, the dynamics of the metapopulation is still not very clear to me. The sentence: "Patches can exist in two states, saturated or empty" is misleading. There is an establishment time where populations are in an intermediate situation. Besides when you write "Saturated patches are occupied by hosts that dominate the local resources, and cannot be invaded" remains unclear. Phage and bacteria are going extinct and recolonise at the same rate? Why? I know you relax this in the Sup Info but you should discuss this sooner.

We revised the text to address this point on lines 113-116 to state: "Patch clearing events, which occur with rate c per patch per unit time, clear a patch of its inhabitants and enable invasion by strains from other patches." We removed any terminology related to saturated or empty patches, including both sentences mentioned by the reviewer. The reference to independent migration of phage is appropriate where currently placed in the text at line 277.

If the main result is to say that spatial structure promotes the evolution of altruistic resistance you should discuss earlier models about this (e.g. Débarre et al 2012 Am Nat 179(1) 52-63,

Fukuyo et al 2012 Sci Rep 2, 238, Berngruber et al 2013 Ecol Letters, 16: 446-453). Discussing these earlier models would help see what is driving the evolution of leaky resistance. In particular no need to introduce the notion of "ecological memory" (what does it mean?) to explain the effect of spatial structure.

We have added the recommended references to altruistic resistance, together with the following statement at line 310: "Additional applications may be relevant in abortive immune systems, where spatial structure has been shown to promote evolution of altruistic resistance mechanisms⁴⁵⁻⁴⁷". The concept of ecological memory captures a fundamental point of our paper, namely that the maintenance of both pathogens and defense mechanisms is explained by dynamics that emerge at the ecological level, rather than at lower levels of organization. This point is discussed starting on line 289, and is unrelated to altruistic resistance, hence it must occur earlier in the manuscript.

We would like to thank the reviewer for their helpful comments, and for their time and effort in reviewing our manuscript.

Reviewer #3 (Remarks to the Author):

The authors have adequately addressed my concerns. The addition of S12 Text describing a model with independently dispersing phage is a nice addition. I think this careful and comprehensive theoretical work makes an important contribution to the phage-host literature.

We would like to thank the reviewer for their helpful comments, and for their time and effort in reviewing our manuscript.